# THE POWER OF FEEL-GOOD THOMPSON SAMPLING: A UNIFIED FRAMEWORK FOR LINEAR BANDITS

## ABSTRACT

Linear contextual bandit is one of the most popular models in online decision-making with bandit feedback. Prior work has studied different variants of this model, e.g., misspecified, non-stationary, and multi-task/life-long linear contextual bandits. However, there is no single framework that can unify the algorithm design and analysis for these variants. In this paper, we propose a unified framework for linear contextual bandits based on feel-good Thompson sampling (Zhang, 2021). The algorithm derived from our framework achieves nearly minimax optimal regret in various settings and resolves the respective open problem in each setting. Specifically, let $d$ be the dimension of the context and $T$ be the length of the horizon, our algorithm achieves an $\widetilde{\mathcal{O}}(d\sqrt{ST})$ regret bound for non-stationary linear bandits with at most $S$ switches, $\widetilde{\mathcal{O}}(d^{\frac{5}{6}}T^{\frac{2}{3}}P^{\frac{1}{3}})$ regret for non-stationary linear bandits with bounded path length $P$, and $\widetilde{\mathcal{O}}(d\sqrt{kT} + \sqrt{dkMT})$ regret for (generalized) lifelong linear bandits over $M$ tasks that share an unknown representation of dimension $k$. We believe our framework will shed light on the design and analysis of other linear contextual bandit variants.

## 1 INTRODUCTION

Linear contextual bandit is one of the most popular models in online decision-making with a large, possibly infinite, action space. This bandit model has been widely studied in the past decade. One of the most successful approaches is based on the upper confidence bound (Auer, 2002). For example, LinUCB (Li et al., 2010) (or OFUL (Abbasi-Yadkori et al., 2011)) follows the optimism-in-the-face-of-uncertainty principle and chooses the best action within an elliptical confidence ball. The algorithm has been proved to be nearly minimax optimal by using the elliptical potential lemma to track the bonus term. With some modifications to the algorithm, one had generalized this algorithm to various settings, e.g., non-stationary linear bandits (Chen et al., 2019), multi-task linear bandits (Hu et al., 2021), to mention a few. The analyses for these generalizations require the corresponding modified elliptical potential lemma, which is, however, hard to derive in general. (One may refer to the technical note (Faury et al., 2021) which discusses the faults in the elliptical potential lemma for non-stationary linear bandits).

Another common approach for online decision-making is exponentially weighted sampling. By sampling from a distribution over actions based on their historical rewards, it gives rise to near-optimal policy-based algorithms for various settings such as the hedge algorithm (Littlestone & Warmuth, 1994) for prediction with expert advice. For contextual bandits, EXP4 (Auer et al., 2002) enjoys a regret bound of

$$\mathbb{E}[\text{Regret}(T)] \leq \mathcal{O}(\sqrt{KT \log |\mathcal{H}|}),$$

where $K$ is the number of actions, $T$ is the length of the horizon, and $\mathcal{H}$ is the feasible policy set. Note that contextual policy-based algorithms usually allow the policy to take round index as context. This gives a natural way to deal with non-stationary environment. For instance, one can solve non-stationary expert problems using meta-experts by following different experts in different rounds (Herbster & Warmuth, 2004). With this idea, one can obtain the regret bound for a variety of bandit models by counting the number of policies $|\mathcal{H}|$, which is easy to do in general.

This motivates us to find a policy-based algorithm for linear contextual bandits. We note that EXP4 is not suitable for our purpose since its regret suffers a polynomial dependence on the number of actions, which can be unbounded in linear contextual bandits. This is due to the fact that EXP4 is designed for general reward functions and does not leverage the linear structure of linear bandits. Given this observation, we raise the following question:

**Can we design an EXP4-type algorithm for linear contextual bandits?**

In this paper, we answer the above question affirmatively. In detail, we propose **F**eel-**G**ood **T**hompson **S**ampling over **L**inear **P**olicies (FGTS.LP), which is a policy-based algorithm for linear contextual bandits. Our algorithm can be regarded as a policy-based adaption of feel-good Thompson sampling (Zhang, 2021) to linear bandits, while this adaption is nontrivial. Our algorithm enjoys a regret bound that is logarithmically dependent on the number of policies and polynomially dependent on the dimension of contexts. To be specific, we prove the following regret bound for FGTS.LP:

**Theorem 1.1** (Regret Bound of FGTS.LP (informal))**.** *Let $d$ be the dimension of the context, $T$ be the length of the horizon, and $\mathcal{H}$ be the set of all feasible policy hypotheses. The regret of FGTS.LP is bounded by*

$$\mathbb{E}[\mathrm{Regret}(T)] \leq \mathcal{O}(\sqrt{dT \log N(\mathcal{H}, \epsilon)} + T\sqrt{d}\epsilon),$$

*where $\epsilon$ is some hyperparameter and $N(\mathcal{H}, \epsilon)$ is the covering number of policy set which contains an $\epsilon$-optimal policy.*

The above theorem provides a general interface to analyze the performance of FGTS.LP in different settings. Following the idea of including round index, FGTS.LP can deal with various linear contextual bandits. The results are highlighted as follows:

**Theorem 1.2** (Regret Bound over Variants of Linear Bandits (informal))**.** *With specific modification for each setting, the regret of FGTS.LP is bounded as*

- $\mathbb{E}[\mathrm{Regret}(T)] \leq \widetilde{\mathcal{O}}(d\sqrt{T} + T\sqrt{d}\zeta)$ *for $\zeta$-misspecified linear contextual bandits.*

- $\mathbb{E}[\mathrm{Regret}(T)] \leq \widetilde{\mathcal{O}}(d\sqrt{ST})$ *for non-stationary linear contextual bandits with at most $S$ switches.*

- $\mathbb{E}[\mathrm{Regret}(T)] \leq \widetilde{\mathcal{O}}(d^{\frac{5}{6}}T^{\frac{2}{3}}P^{\frac{1}{3}})$ *for non-stationary linear contextual bandits with path length bounded by $P$.*

- $\mathbb{E}[\mathrm{Regret}(T)] \leq \widetilde{\mathcal{O}}(d\sqrt{kT} + \sqrt{dkMT})$ *for (generalized) lifelong linear contextual bandits over $M$ tasks that share an unknown representation of dimension $k$.*

We note that the above results are all near-optimal which match or improve the state-of-the-art in the corresponding settings. To sum up, our contributions are:

- We propose a unified framework for design and analyze various linear contextual bandit models. Our framework is easy to interpret and enjoys near-optimal regret bound in different settings.

- We propose the first nearly minimax algorithm for non-stationary linear contextual bandits with a bounded number of switches.

- We propose a new algorithm for non-stationary linear contextual bandits with bounded path length. It is the first algorithm that achieves nearly minimax regret.

- We propose the first near-optimal algorithm for (generalized) lifelong linear contextual bandits. Its regret matches the state-of-the-art for multi-task linear contextual bandits (Hu et al., 2021), which is a special case of our model.

**Notation.** We use lower and upper case bold face letters to denote vectors and matrices respectively. We use $[k]$ to denote the set $\{1, 2, \cdots, k\}$. We denote the Euclidean norm of vector $\mathbf{x} \in \mathbb{R}^d$ by $\|\mathbf{x}\|_2$. For a matrix $\mathbf{A} = [\mathbf{a}_1, \cdots, \mathbf{a}_k] \in \mathbb{R}^{d \times k}$, we define $\|\mathbf{A}\|_{2,\infty} = \max_{1 \leq i \leq k} \|\mathbf{a}_i\|_2$. For two non-negative sequence $\{a_n\}, \{b_n\}$, we write $a_n \leq \mathcal{O}(b_n)$ if there exists an absolute constant $C > 0$ such that $a_n \leq Cb_n$ for all $n \geq 1$, and $a_n \leq \widetilde{\mathcal{O}}(b_n)$ if there exists an absolute constant $k$ such that $a_n \leq \mathcal{O}(b_n \log^k b_n)$; we write $a_n \geq \Omega(b_n)$ if there exists an absolute constant $C > 0$ such that $a_n \geq Cb_n$ for all $n \geq 1$ and $a_n \geq \widetilde{\Omega}(b_n)$ if there exists absolute constant $k$ such that $a_n \geq \Omega(b_n \log^{-k} b_n)$; we write $a_n = \Theta(b_n)$ if there exists absolute constants $0 < C_1 \leq C_2$ such that $C_1 b_n \leq a_n \leq C_2 b_n$ for all $n \geq 1$. For any set $\mathcal{C}$, we use $|\mathcal{C}|$ to denote its cardinality. We use $\log$ to denote $\log_e$ for short.

## 2 RELATED WORK

**Misspecified Linear Bandits.** The misspecified linear bandits was first studied by Ghosh et al. (2017), and they proposed an algorithm that achieves sub-linear regret when the misspecification $\zeta$ is small. Lattimore & Szepesvari (2020) proposed an algorithm with an $\widetilde{\mathcal{O}}(d\sqrt{T} + T\sqrt{d}\zeta)$ regret

but requiring the contexts to be stationary. The regret is nearly minimax optimal as they proved a matching lower bound in the paper. Later, Zanette et al. (2020) proposed a LinUCB-like algorithm with the same regret that gets rid of the requirement of stationary context. More recent works (Foster et al., 2020a; Krishnamurthy et al., 2021; Takemura et al., 2021) studied the problem when the misspecification level is unknown.

**Non-Stationary Linear Bandits with Bounded Switches.** Non-stationary online decision-making models with bounded switches have long been studied in the literature. Cesa-Bianchi et al. (1993) studied the problem in the full information experts setting, and proposed an algorithm with an expected regret bound $\widetilde{\mathcal{O}}(\sqrt{ST})$, where $S$ is the number of switches. Later, a high probability bound was obtained by a a UCB-type algorithm (Auer, 2002). In the bandit feedback setting, EXP3.S (Auer et al., 2001) achieves a regret of $\widetilde{\mathcal{O}}(\sqrt{KST})$, where $K$ is the number of actions. Borrowing the idea from EXP4, the algorithm has been extended to the contextual setting with the $\widetilde{\mathcal{O}}(\sqrt{KST \log |\Pi|})$ regret bound (Luo et al., 2018), where $\Pi$ is the finite policy class. More recently, Luo et al. (2022) studied the non-stationary linear bandit setting where the action set is a unit ball and proposed a bandit-over-bandit approach that achieves a regret of $\widetilde{\mathcal{O}}(\sqrt{dST})$. There is no existing algorithms for non-stationary linear contextual bandits with bounded switches.

**Non-Stationary Linear Bandits with Bounded Path Length.** In recent years, non-Stationary linear bandits with bounded path length have received increasing attention. Various algorithms have been developed (Cheung et al., 2019; Chen et al., 2019; Zhao et al., 2020) in this setting. The key idea behind these algorithms is to progressively forget the past data, and the proof is based on the same kind of elliptical potential lemma. However, as discussed by Faury et al. (2021), there is a fault in the proof of the original paper (Cheung et al., 2019), and the corrected analysis can only get a regret bound of $\widetilde{\mathcal{O}}(d^{\frac{3}{4}} T^{\frac{3}{4}} P^{\frac{1}{4}})$. Meanwhile, the lower bound of this problem is $\Omega(T^{\frac{2}{3}} P^{\frac{1}{3}})$ (Faury et al., 2021). The algorithm derived from our framework closes the gap on $T$.

**(Generalized) Lifelong Linear Bandits with Shared Representation.** The multi-task linear bandit, in which the agent plays over a collection of tasks simultaneously, was first studied by Yang et al. (2020). They proposed an explore-then-commit algorithm that leverages the multi-task structure but requires a unit-ball action space. Their followed-up work (Yang et al., 2022) improved the regret of the algorithm and generalized it to the lifelong setting. The algorithm has a regret bound of $\widetilde{\mathcal{O}}(d\sqrt{kT} + k\sqrt{MT})$. Hu et al. (2021) studied the multi-task setting in linear contextual bandits and proposed a computationally inefficient algorithm with a regret of $\widetilde{\mathcal{O}}(d\sqrt{kT} + \sqrt{dkMT})$ using LinUCB-type approach. Recent work (Qin et al., 2022) studied the non-stationary lifelong linear bandits with a task diversity assumption. On the lower bound side, Yang et al. (2020) proved that any algorithm for multi-task linear bandits suffers at least an $\Omega(d\sqrt{kT} + k\sqrt{MT})$ regret.

## 3 THE PROPOSED FRAMEWORK

### 3.1 PROBLEM SETUP

We first introduce a framework which is able to cover variants of linear contextual bandits. Let $d$ be the dimension of the context, and $T$ be the lenght of horizon. Denote by $\mathcal{A} \subseteq \{\mathbf{a} \in \mathbb{R}^d : \|\mathbf{a}\|_2 \leq 1\}$ the action space and $\mathcal{F} \subseteq \mathbb{R}^{\mathcal{A}}$ the reward function space. The contextual bandit can be described as a repeated game between an agent (the bandit algorithm) and the environment (the adversary). In each round $t = 1, \cdots, T$, the environment first picks a hidden reward function $f^{(t)} \in \mathcal{F}$ and draws an action set $\mathcal{A}^{(t)} \subseteq \mathcal{A}$. After observing the action set $\mathcal{A}^{(t)}$, the agent selects an action $\mathbf{a}^{(t)} \in \mathcal{A}^{(t)}$ and receives a stochastic reward

$$r^{(t)} = f^{(t)}(\mathbf{a}^{(t)}) + \xi^{(t)},$$

where $f^{(t)}(\mathbf{a}^{(t)})$ is the expected value of the observed reward $r^{(t)}$ and $\xi^{(t)}$ is a zero mean random noise satisfying $\mathbb{E}[\xi^{(t)}|\Omega^{(t-1)}, f^{(t)}, \mathcal{A}^{(t)}, \mathbf{a}^{(t)}] = 0$, where $\Omega^{(t)} = \{(f^{(\tau)}, \mathcal{A}^{(t)}, \mathbf{a}^{(\tau)}, r^{(\tau)})\}_{\tau=1}^{t}$ is the history in the first $t$ rounds. The learning objective of the agent is to maximize the expected cumulative reward, or equivalently, to minimize the pseudo-regret

$$\text{Regret}(T) := \sum_{t=1}^{T} \Big( f^{(t)}(\mathbf{a}_*^{(t)}) - f^{(t)}(\mathbf{a}^{(t)}) \Big), \tag{1}$$

where $\mathbf{a}_*^{(t)} = \arg\max_{\mathbf{a} \in \mathcal{A}^{(t)}} f^{(t)}(\mathbf{a})$ is the optimal action in round $t$. We assume the reward functions can be approximated by linear functions.

**Assumption 3.1.** *There is a mapping* $\boldsymbol{\theta} : \mathcal{F} \to \{\mathbf{x} \in \mathbb{R}^d : \|\mathbf{x}\|_2 \le 1\}$ *and a universal constant* $\zeta \in [0, 1]$ *such that*

$$\sup_{(f, \mathbf{a}) \in (\mathcal{F} \times \mathcal{A})} |\langle \mathbf{a}, \boldsymbol{\theta}(f) \rangle - f(\mathbf{a})| \le \zeta.$$

*Moreover, we assume* $|f(\mathbf{a})| \le 1$ *for all* $(f, \mathbf{a}) \in (\mathcal{F} \times \mathcal{A})$.

We call $\zeta$ the misspecification level of the model. For the special case $\zeta = 0$, the model reduces to standard linear bandits. We further assume the observed reward is universally bounded.

**Assumption 3.2.** *The observed reward always satisfies* $|r^{(t)}| \le 1$ *for all* $t \in [T]$.

This assumption ensures the additive noise $\xi^{(t)}$ is bounded. Note that this assumption is not essential to the algorithm, our analysis can be naturally generalized to unbounded sub-Gaussian with constant variance as it showed by Zhang (2021). We use bounded noise to avoid introducing variance as an extra parameter.

The above framework is compatible with variants of linear contextual bandits. For example, one can formulate specific linear contextual bandits by restricting the environment to select reward sequences $(f^{(1)}, \cdots, f^{(T)})$ from the corresponding structural set. We will present some concrete instances in the sequel.

### 3.2 SPECIFIC EXAMPLES

#### 3.2.1 STATIONARY MISSPECIFIED LINEAR BANDITS

For stationary linear contextual bandits, the reward function is fixed before agent decides. Thus, the environment is restricted to select reward function sequences that has the same linear approximation across rounds.

**Assumption 3.3.** *There exists a vector* $\boldsymbol{\theta}_0 \in \mathbb{R}^d$ *such that* $\boldsymbol{\theta}(f^{(t)}) = \boldsymbol{\theta}_0$ *for all* $t \in [T]$.

The reward function is allowed to be misspecified from linear functions in Assumption 3.1. It is easy to verify that our framework reduces to linear contextual bandits with misspecification (Foster et al., 2020a) under Assumptions 3.1- 3.3.

#### 3.2.2 NON-STATIONARY LINEAR BANDITS WITH BOUNDED SWITCHES

For non-stationary linear bandits, we first consider the case where the reward function may change dramatically for a finite number of times (Auer et al., 2001; Luo et al., 2022). The agent is not told when and how the reward function switches, and the environment can schedule the change in an adversarial way. We describe this setting using the following assumption:

**Assumption 3.4.** *There exists a constant* $S$ *such that* $\sum_{t=2}^T \mathbb{1}[\boldsymbol{\theta}(f^{(t)}) \ne \boldsymbol{\theta}(f^{(t-1)})] \le S$.

We call $S$ the number of switches. We call the problem formulated under Assumptions 3.1, 3.2 and 3.4 as the non-stationary linear bandits with bounded switches. We note that the non-stationary linear bandits with bounded switches is a harder problem than that with bounded path length, since there is a black-box reduction from non-stationary linear bandits with bounded path length to non-stationary misspecified linear bandits with bounded switches. Recently, Luo et al. (2022) studied this setting but for non-contextual linear bandits with $\mathcal{A}^{(t)} = \{\mathbf{a} \in \mathbb{R}^d : \|\mathbf{a}\|_2 \le 1\}$. They proposed an algorithm with $T^{1/2}$ regret. However, there is no existing algorithm for non-stationary contextual linear bandits with $T^{1/2}$ regret.

#### 3.2.3 NON-STATIONARY LINEAR BANDITS WITH BOUNDED PATH LENGTH

We also consider another kind of non-stationary linear bandits where the reward function can drift slowly over time (Cheung et al., 2019). The agent does not know the evolution dynamic, while the environment can choose the reward function adversarially in each round. We characterize this setting by the following assumption.

**Assumption 3.5.** *There exists a constant* $P$ *such that* $\sum_{t=2}^T \|\boldsymbol{\theta}(f^{(t)}) - \boldsymbol{\theta}(f^{(t-1)})\|_2 \le P$.

We call $P$ the path length. One can verify our framework under Assumptions 3.1, 3.2 and 3.5 reduces to non-stationary linear bandits with bounded path length (Faury et al., 2021). Existing regret upper bound is of order $T^{3/4}$ while the lower bound is of order $T^{2/3}$. So there is still a gap between the regret upper and lower bounds.

### 3.2.4 (GENERALIZED) LIFELONG LINEAR BANDITS WITH SHARED REPRESENTATION

We consider the setting where the agents have to solve a collection of correlated tasks sequentially. Let $M$ be the number of tasks. At the beginning of the process, the environment first chooses $M$ tasks $f_1, \cdots, f_M$. In each round $t$, the environment draws a task index $m^{(t)} \in [M]$ and assign current reward function as $f^{(t)} = f_{m^{(t)}}$. Besides the action set $\mathcal{A}^{(t)}$, the agent is provided with the task index $m^{(t)}$ and selects the action $\mathbf{a}^{(t)}$.

It is worth noting that our lifelong learning setting is a generalized version of the lifelong learning setting studied in Yang et al. (2022). The original setting of lifelong learning, in which the same task only appears in one interval, is equivalent to the case where the same $m^{(t)}$ lies in a single interval. We consider this generalized setting since it also captures multi-task learning (Hu et al., 2021), where the tasks appear periodically. We make the following assumption.

**Assumption 3.6.** *There exists a hidden orthogonal matrix $\mathbf{B} \in \mathbb{R}^{d \times k}$ and a set of hidden vectors $\{\mathbf{w}_i\}_{i=1}^{M}$ with $\mathbf{w}_i \in \mathbb{R}^k$ such that $\boldsymbol{\theta}(f^{(t)}) = \mathbf{B}\mathbf{w}_{m^{(t)}}$ holds for all $t \in [T]$ where $\{m^{(t)}\}_{t=1}^{T}$ is some sequence with $m^{(t)} \in [M]$.*

In Assumption 3.6, $\mathbf{B}$ is called the linear feature extractor of the model. The assumption implies $[\boldsymbol{\theta}(f^{(1)}), \cdots, \boldsymbol{\theta}(f^{(T)})]$ is a low rank matrix which can be decomposed over the bases $\mathbf{B}$. This is a common assumption for representation learning over linear bandits in the literature (see e.g., (Hu et al., 2021; Yang et al., 2022)). The genearlized lifelong linear bandits is defined under our framework with Assumptions 3.1, 3.2 and 3.6. Recently, Yang et al. (2022) studied the lifelong learning setting for non-contextual linear bandits where $\mathcal{A}^{(t)} = \{\mathbf{a} \in \mathbb{R}^d : \|\mathbf{a}\|_2 \leq 1\}$ and proposed an algorithm with $T^{1/2}$ regret. However, there is no existing algorithm for lifelong contextual linear bandits with $T^{1/2}$ regret.

## 4 FEEL-GOOD THOMPSON SAMPLING OVER LINEAR POLICIES

In this section, we present our main algorithm FGTS.LP (**F**eel-**G**ood **T**hompson **S**ampling over **L**inear **P**olicies) for solving linear context bandits under the framework described in Section 3. We first introduce some important concepts in the algorithm design.

### 4.1 LINEAR POLICY CLASS AND ITS COVERING

It is pretty annoying that the regret defined in (1) is related with dynamic variable $\mathcal{A}^{(t)}$. We introduce an essential concept in our algorithm. Instead of selecting an action $\mathbf{a}^{(t)}$, we let the bandit algorithm to select a sequence of policies $\pi^{(t)}$, which decide how the agent take action based on the observed context over time. Formally, a sequence of policies $\pi : [T] \times \mathcal{A} \to \mathbb{R}$ is a time-variant hypothesis for the expected reward of choosing any actions in any round, i.e., the expected reward of action $\mathbf{a} \in \mathcal{A}$ on round $t$ is $\pi(t, \mathbf{a})$. Thus, the sequence suggests to play action $\mathbf{a}^{(t)} = \arg\max_{\mathbf{a} \in \mathcal{A}^{(t)}} \pi(t, \mathbf{a})$ when the action set $\mathcal{A}^{(t)}$ is realized on round $t$. Since the reward functions in our setting can be approximated by linear functions, we would focus on linear policies that can be approximated by linear functions:

**Definition 4.1** (Linear Policy). *A linear policy $\pi(\boldsymbol{\Theta})$ is a sequence of models parameterized by the matrix $\boldsymbol{\Theta} = [\boldsymbol{\theta}_1, \cdots, \boldsymbol{\theta}_T] \in \mathbb{R}^{d \times T}$ such that $\pi(\boldsymbol{\Theta})(t, \mathbf{a}) = \langle \mathbf{a}, \boldsymbol{\theta}_t \rangle$.*

Let $\mathcal{H}$ be the set of all feasible linear policies. Under certain assumptions within our framework, the linear policy set is restricted to a corresponding domain. For example, under Assumption 3.3, the linear policy set $\mathcal{H}$ is parameterized by matrices $\boldsymbol{\Theta}$ in which all columns are the same.

Note that $\mathcal{H}$ is a set over continuous domain and thus of infinite size. To deal with the infinite number of linear policies, we introduce the approach of $\epsilon$-net and covering number.

**Definition 4.2** ($\epsilon$-Net of Linear Policies). *$\mathcal{H}_\epsilon$ is said to be a $\epsilon$-net of $\mathcal{H}$ if any linear policy $\boldsymbol{\Theta} = [\boldsymbol{\theta}_1, \cdots, \boldsymbol{\theta}_T] \in \mathcal{H}$ has an $\epsilon$-approximation $\widetilde{\boldsymbol{\Theta}} = [\widetilde{\boldsymbol{\theta}}_1, \cdots, \widetilde{\boldsymbol{\theta}}_T] \in \mathcal{H}_\epsilon$ with $\|\boldsymbol{\theta}_t - \widetilde{\boldsymbol{\theta}}_t\|_2 \leq \epsilon$ for all $t \in [T]$. Moreover, we define the covering number of $\mathcal{H}$ (at scale $\epsilon$) be the size of minimum $\mathcal{H}_\epsilon$, i.e., $N(\mathcal{H}, \epsilon, \|\cdot\|_{2,\infty}) := \inf |\mathcal{H}_\epsilon|$.*

One can regard the metric entropy $\log N(\mathcal{H}, \epsilon, \|\cdot\|_{2,\infty})$ as the effective dimension of the policy set.

### 4.2 THE ALGORITHM

Now we are ready to present our main algorithm. The algorithm is an adaptation of the FGTS (Zhang, 2021). We present the pseudo-code in Algorithm 1.

---

**Algorithm 1** FGTS.LP

---

**input:** linear policies set $\mathcal{H}$; inverse temperature $\beta$; exploration parameter $\lambda$; covering radius $\epsilon$.
construct an $\epsilon$-net $\mathcal{H}_\epsilon$ of $\mathcal{H}$ such that $\log |\mathcal{H}_\epsilon| = \mathcal{O}(\log N(\mathcal{H}, \epsilon, \|\cdot\|_{2,\infty}))$
**for** round $t = 1, \cdots, T$ **do**
    sample $\boldsymbol{\Theta}^{(t)} = [\boldsymbol{\theta}_1^{(t)}, \cdots, \boldsymbol{\theta}_T^{(t)}] \sim P^{(t)}(\boldsymbol{\Theta}) \propto \exp\big(-\beta \sum_{\tau=1}^{t-1} L^{(\tau)}(\boldsymbol{\Theta})\big)$ for (3) over $\mathcal{H}_\epsilon$
    take action $\mathbf{a}^{(t)} \leftarrow \arg\max_{\mathbf{a} \in \mathcal{A}^{(t)}} \langle \mathbf{a}, \boldsymbol{\theta}_t^{(t)} \rangle$
**end for**

---

At the beginning, the algorithm constructs an $\epsilon$-net $\mathcal{H}_\epsilon$ over all feasible linear policies $\mathcal{H}$. In each round $t$, the algorithm first samples a policy $\boldsymbol{\Theta}^{(t)}$ over $\mathcal{H}_\epsilon$ according to the distribution

$$P^{(t)}(\boldsymbol{\Theta}) := \frac{\exp\big(-\beta \sum_{\tau=1}^{t-1} L^{(\tau)}(\boldsymbol{\Theta})\big)}{\sum_{\boldsymbol{\Theta}' \in \mathcal{H}_\epsilon} \exp\big(-\beta \sum_{\tau=1}^{t-1} L^{(\tau)}(\boldsymbol{\Theta}')\big)}, \tag{2}$$

where $\beta > 0$ is the inverse temperature and $L^{(t)} : \mathbb{R}^{d \times T} \to \mathbb{R}$ is a loss function defined as follows

$$L^{(t)}(\boldsymbol{\Theta}) = (r^{(t)} - \langle \mathbf{a}^{(t)}, \boldsymbol{\theta}_t \rangle)^2 - \lambda \max_{\mathbf{a} \in \mathcal{A}^{(t)}} \langle \mathbf{a}, \boldsymbol{\theta}_t \rangle. \tag{3}$$

Here $\lambda > 0$ is a tuning parameter which controls the exploration. The algorithm then chooses the optimal action $\mathbf{a}^{(t)}$ that maximizes the expected reward according to the chosen policy $\boldsymbol{\Theta}^{(t)}$ among the action set $\mathcal{A}^{(t)}$.

We note the loss function in (3) has two terms. The first term is the Thompson sampling term. It casts penalty based on the estimation error of the reward. Recall $r^{(\tau)}$ is a random variable with mean $f^{(\tau)}(\mathbf{a}^{(\tau)}) \approx \langle \mathbf{a}^{(\tau)}, \boldsymbol{\theta}(f^{(\tau)}) \rangle$, this term forces the algorithm to select policy close to the true policy $\boldsymbol{\theta}(f^{(\tau)})$. This encourages the algorithm to do exploitation. The second term is so called feel-good exploration term (Zhang, 2021). It favors the policy that grants large rewards for historical actions, which can be regarded as an bonus that encourages the algorithm to do exploration.

Note that Algorithm 1 needs to know $T$ and $\mathcal{H}$ before hand. By restarting the algorithm periodically with the doubling trick, one can run the algorithm without knowing $T$ in hindsight. However, it is unclear if the algorithm can be parameter-free for $\mathcal{H}$ since model selection for exponential weighted sampling algorithm remains an open question (Foster et al., 2020b). Furthermore, the algorithm is sampled over $\epsilon$-net of $\mathcal{H}$, which is not necessarily computationally efficient in practice. So it is better to regard Algorithm 1 as an oracle-efficient algorithm provided the sampling oracle over $\mathcal{H}_\epsilon$. For the implementation of this algorithm in practice, we present a detailed discussion in Appendix B

### 4.3 COMPARISON WITH EXISTING ALGORITHMS

**Comparison with FGTS.** One may see that FGTS.LP has the same structure as the original FGTS (Zhang, 2021). Both algorithms use exponentially weighted sampling over the same loss function. The main difference between the two algorithms is that the original FGTS samples over models (or equivalently, reward functions) rather than policies. Note that sampling over models is equivalent to choosing the stationary policies, which always select the same action given the same context. Thus, the original FGTS cannot deal with non-stationary environments in which the reward functions change over time. In contrast, sampling over policies enables us to inspect the misspecification explicitly and thus analyze it meticulously. As a result, FGTS.LP can achieve nearly minimax regret bound for misspecified linear bandits and non-stationary linear bandits.

**Comparison with EXP4.** Another contextual bandit algorithm that uses exponentially weighted sampling is EXP4 (Auer et al., 2002). Using exponentially weighted sampling, one can corral a large set of policies achieving a low regret as long as the reward for each policy is revealed. However, the agent cannot observe the payoff for policies that are not chosen in the bandits setting, and has to estimate it. EXP4 uses an unbiased estimator to estimate the reward. However, the estimator has a large variance which is proportional to the number of actions. This causes a polynomial dependence on the size of the action set in the regret bound. As a comparison, FGTS.LP uses the expected reward reported by the policies but casts penalties on the policies that cannot estimate the payoff of chosen actions. Although this estimator may be biased in estimating the actual reward, the estimator will assigns a large value to the policy with a high return. So the regret analysis for exponentially weighted sampling (Littlestone & Warmuth, 1994) can still be applied to FGTS.LP. Since the variance of this estimator is much smaller and independent of the number of actions, FGTS.LP is able to get a regret bound that is independent of the size of the action set.

**Comparison with LinUCB.** In addition to Thompson sampling and posterior sampling, another common approach for linear contextual bandits is LinUCB (Li et al., 2010) (or OFUL (Abbasi-Yadkori et al., 2011)). The algorithm implements the optimism-in-face-of-uncertainty principle, which selects the policy with the largest optimistic reward among all possible policies that agree with the observed reward in history. Note that FGTS.LP tends to sample policy with large estimated reward on the historical data and small estimation error. Therefore, FGTS.LP can also be regarded as an implementation of the same principle. The difference is, rather than focusing on one action, FGTS.LP uses exponentially weighted sampling to select a policy. From the game theory perspective, the variants of linear contextual bandits can be regarded as a two-player game. It is known that a mixed strategy profile usually generates better rewards than a pure strategy profile against adversarial opponents (Fudenberg & Tirole, 1991). Thus, FGTS.LP will have a stronger ability to solve variants of linear contextual bandits.

## 5    MAIN RESULTS

In this section, we present the theoretical results of our algorithm. The following theorem provides a regret guarantee for Algorithm 1 when applied to general linear bandits.

**Theorem 5.1.** *Suppose $\epsilon \in [0, 1]$, set $\lambda = \Theta\big(\sqrt{\log N(\mathcal{H}, \epsilon, \|\cdot\|_{2,\infty})/(dT) + (\epsilon + \zeta)^2/d}\big)$ and $\beta = \Theta(1)$. Under Assumptions 3.1 and 3.2, for any policy class $\mathcal{H}$, the expected regret of Algorithm 1 is bounded by*

$$\mathbb{E}[\text{Regret}(T)] \leq \mathcal{O}\Big(\sqrt{dT \log N(\mathcal{H}, \epsilon, \|\cdot\|_{2,\infty})} + T\sqrt{d}(\epsilon + \zeta)\Big), \tag{4}$$

*where the expectation is taken over all randomness of the learning algorithm and the data noise.*

There are two terms on the RHS of (4), which we will explain separately. The first term depends on the metric entropy $\log N(\mathcal{H}, \epsilon, \|\cdot\|_{2,\infty})$, which depicts the complexity of the policy set $\mathcal{H}$. The second term depends on the scale of the $\epsilon$-net $\mathcal{H}_\epsilon$ as well as the misspecification level $\zeta$. There is a trade-off between the first term and the second term driven by the scale of the $\epsilon$-net: for large $\epsilon$, the first term is small and the second term is large; for small $\epsilon$, the first term will be large and the second term will be small.

With Theorem 5.1, one only needs to calculate the metric entropy $\log N(\mathcal{H}, \epsilon, \|\cdot\|_{2,\infty})$ and select proper $\epsilon$ to minimize (4), i.e., ensuring $\log N(\mathcal{H}, \epsilon, \|\cdot\|_{2,\infty}) \sim \sqrt{T}(\epsilon + \zeta)$. Since the metric entropy $\log N(\mathcal{H}, \epsilon, \|\cdot\|_{2,\infty})$ is the effective dimension of the policy class, it is easy to calculate for different policy classes and derive the corresponding regret bound. As a result, Algorithm 1 can be viewed as a unified framework for linear contextual bandits.

### 5.1    IMPLICATIONS TO LINEAR BANDIT VARIANTS

In this subsection, we show the implications of Theorem 5.1 on the specific examples of contextual linear bandit variants. For the ease of comparison between our results and prior results, we summarize the results in Table 1.

#### 5.1.1    MISSPECIFIED LINEAR BANDITS

We start with the simple setting where the approximate linear function is stationary. In this setting, the $\epsilon$-net of the linear policy class can be reduced to the $\epsilon$-net of single reward function. Since the embedding is of dimension $d$, the covering number is exactly $\mathcal{O}(\epsilon^{-d})$:

**Lemma 5.1** (Covering Number for Misspecified Linear Bandits)**.** *Under Assumptions 3.1 and 3.3, the metric entropy of linear policy class satisfies*

$$\log N(\mathcal{H}, \epsilon, \|\cdot\|_{2,\infty}) \leq \mathcal{O}(d \log \epsilon^{-1}).$$

By choosing $\epsilon = T^{-1}$, the metric entropy can be bounded as $\log N(\mathcal{H}, \epsilon, \|\cdot\|_{2,\infty}) \leq \mathcal{O}(d \log T)$. With Theorem 5.1, we directly get the following regret bound:

**Corollary 5.1** (Upper Bound for Misspecified Linear Bandits)**.** *Under Assumptions 3.1, 3.2 and 3.3, the expected regret of FGTS.LP is bounded by*

$$\mathbb{E}[\text{Regret}(T)] \leq \mathcal{O}\Big(d\sqrt{T \log T} + T\sqrt{d}\zeta\Big).$$

The following lower bound shows that the regret upper bound in Corollary 5.1 is tight up to logarithmic factors:

| | Existing Algorithm | FGTS.LP (This Paper) | Lower Bounds |
|---|---|---|---|
| Misspecified LCB | $\widetilde{\mathcal{O}}(d\sqrt{T} + T\sqrt{d}\zeta)$ (Zanette et al., 2020) | $\widetilde{\mathcal{O}}(d\sqrt{T} + T\sqrt{d}\zeta)$ (Corollary 5.1) | $\widetilde{\Omega}(d\sqrt{T} + T\sqrt{d}\zeta)$ (Lattimore & Szepesvari, 2020) |
| Non-stationary LCB (Bounded Switches) | – | $\widetilde{\mathcal{O}}(d\sqrt{ST})$ (Corollary 5.2) | $\widetilde{\Omega}(d\sqrt{ST})$ (Lemma F.5) |
| Non-stationary LCB (Bounded Length) | $\widetilde{\mathcal{O}}(d^{\frac{3}{4}}T^{\frac{3}{4}}P^{\frac{1}{4}})$ (Faury et al., 2021) | $\widetilde{\mathcal{O}}(d^{\frac{5}{6}}T^{\frac{2}{3}}P^{\frac{1}{3}})$ (Corollary 5.3) | $\widetilde{\Omega}(d^{\frac{2}{3}}T^{\frac{2}{3}}P^{\frac{1}{3}})$ (Cheung et al., 2019) |
| Multi-task LCB | $\widetilde{\mathcal{O}}(d\sqrt{kT} + \sqrt{dkMT})$ (Hu et al., 2021) | $\widetilde{\mathcal{O}}(d\sqrt{kT} + \sqrt{dkMT})$ (Corollary 5.4) | $\widetilde{\Omega}(d\sqrt{kT} + k\sqrt{MT})$ (Yang et al., 2020) |
| Lifelong LCB | – | $\widetilde{\mathcal{O}}(d\sqrt{kT} + \sqrt{dkMT})$ (Corollary 5.4) | $\widetilde{\Omega}(d\sqrt{kT} + k\sqrt{MT})$ (Yang et al., 2020) |

Table 1: Summary of the results on variants of linear contextual bandits. $d$ is the dimension of the context, $T$ is the length of horizon, and $\zeta$ is the misspecification level for misspecified bandits. In addition, $S$ is the number of switches and $P$ is the path length for non-stationary bandits. Furthermore, $M$ denotes the number of tasks and $k$ denotes the representation dimension for multi-task and lifelong bandits.

**Proposition 5.1** (Lower Bound for Misspecified Linear Bandits)**.** *Under Assumptions 3.1, 3.2 and 3.3, for any algorithm, there exists a bandit instance for which*

$$\mathbb{E}[\mathrm{Regret}(T)] \geq \Omega\Big(d\sqrt{T} + T\sqrt{d/\log T}\zeta\Big).$$

### 5.1.2 NON-STATIONARY LINEAR BANDITS WITH BOUNDED SWITCHES

In this setting, we can construct the $\epsilon$-net by enumerating the positions of switches and cover the reward function set after each switch. The set of reward functions after each switch has a covering number of $\mathcal{O}(\epsilon^{-d})$. So the set of reward functions with $S$ switches has a covering number of $\mathcal{O}(\epsilon^{-dS})$. Moreover, the different positions of switches within $T$ rounds can be bounded by $\mathcal{O}(T^S)$. With above facts, we can bound the size of $\epsilon$-net:

**Lemma 5.2** (Covering Number for Non-Stationary Linear Bandits with Bounded Switches)**.** *Under Assumptions 3.1 and 3.4, the metric entropy of the linear policy set satisfies*

$$\log N(\mathcal{H}, \epsilon, \|\cdot\|_{2,\infty}) \leq \mathcal{O}(dS\log\epsilon^{-1} + S\log T).$$

Using Theorem 5.1 with $\epsilon = T^{-1}$, we immediately obtain the following regret bound:

**Corollary 5.2** (Upper Bound for Non-Stationary Linear Bandits with Bounded Switches)**.** *Under Assumptions 3.1, 3.2 and 3.4, the expected regret of FGTS.LP is bounded by*

$$\mathbb{E}[\mathrm{Regret}(T)] \leq \mathcal{O}\Big(d\sqrt{ST\log T} + T\sqrt{d}\zeta\Big).$$

The following lower bound shows FGTS.LP is nearly minimax optimal for non-stationary linear bandits with bounded switches:

**Proposition 5.2** (Lower Bound for Non-Stationary Linear Bandits with Bounded Switches)**.** *Under Assumptions 3.1, 3.2 and 3.4, for any algorithm, there exists a bandit instance for which*

$$\mathbb{E}[\mathrm{Regret}(T)] \geq \Omega\Big(d\sqrt{ST} + T\sqrt{d/\log T}\zeta\Big).$$

### 5.1.3 NON-STATIONARY LINEAR BANDITS WITH BOUNDED PATH LENGTH

Now we move to the more challenging setting where the parameters slowly drift over time. We would reduce the parameter drift to parameter switch. More specifically, a parameter switch is considered to happen only if the path length of parameter drift from last switch exceeds $\epsilon/2$. Since the total path length is bounded by $P$, there are no more than $2P\epsilon^{-1}$ switches. As a result, an $\epsilon/2$-net of the reward functions for non-stationary linear bandits with no more than $2P\epsilon^{-1}$ switches is an $\epsilon$-net of the reward functions for non-stationary linear bandits with path length no more than $P$. This immediately gives the following result on its covering number:

**Lemma 5.3** (Covering Number for Non-Stationary Linear Bandits with Bounded Path Length). *Under Assumptions 3.1 and 3.5, the metric entropy of linear policies satisfies*

$$\log N(\mathcal{H}, \epsilon, \|\cdot\|_{2,\infty}) \leq \mathcal{O}(d \log \epsilon^{-1} + dP\epsilon^{-1} \log \epsilon^{-1} + P\epsilon^{-1} \log T).$$

Using Theorem 5.1 with $\epsilon = \max\{T^{-\frac{1}{3}} d^{\frac{1}{3}} P^{\frac{1}{3}}, T^{-1}\}$, we get the following regret bound:

**Corollary 5.3** (Upper Bound for Non-Stationary Linear Bandits with Bounded Path Length). *Under Assumptions 3.1, 3.2 and 3.5, the expected regret of FGTS.LP is bounded by*

$$\mathbb{E}[\text{Regret}(T)] \leq \widetilde{\mathcal{O}}\Big(d^{\frac{5}{6}} T^{\frac{2}{3}} P^{\frac{1}{3}} + d\sqrt{T} + T\sqrt{d}\zeta\Big).$$

The following lower bound shows that the regret upper bound in Corollary 5.3 is nearly optimal:

**Proposition 5.3** (Lower Bound for Non-Stationary Linear Bandits with Bounded Path Length). *Under Assumptions 3.1, 3.2 and 3.5, for any algorithm, there exists a bandit instance for which*

$$\mathbb{E}[\text{Regret}(T)] \geq \Omega\Big(d^{\frac{2}{3}} T^{\frac{2}{3}} P^{\frac{1}{3}} + d\sqrt{T} + T\sqrt{d/\log T}\zeta\Big).$$

### 5.1.4 (GENERALIZED) LIFELONG LINEAR BANDITS WITH SHARED REPRESENTATION

In this setting, every feasible linear policy can be described by a low-rank matrix that is the product of two matrices $\mathbf{B} \in \mathbb{R}^{d \times k}$ and $[\mathbf{w}_i]_{i=1}^{M} \in \mathbb{R}^{k \times M}$. We can construct the $\epsilon$-net of linear policies using the product of two $\epsilon/2$-net over $\mathbf{B}$ and $[\mathbf{w}_i]_{i=1}^{M}$. Also, the dimension of linear policies is $dk + kM$. This implies the following result on its covering number:

**Lemma 5.4** (Covering Number for Lifelong with Shared Representation). *Under Assumptions 3.1 and 3.6, the metric entropy of linear policies satisfies*

$$\log N(\mathcal{H}, \epsilon, \|\cdot\|_{2,\infty}) \leq \mathcal{O}\big(dk \log \epsilon^{-1} + kM \log \epsilon^{-1}\big).$$

Using Theorem 5.1 with $\epsilon = T^{-1}$, we obtain the following regret bound:

**Corollary 5.4** (Upper Bound for Lifelong Linear Bandits with Shared Representation). *Under Assumptions 3.1, 3.2 and 3.6, the expected regret of FGTS.LP is bounded by*

$$\mathbb{E}[\text{Regret}(T)] \leq \widetilde{\mathcal{O}}\Big(d\sqrt{kT} + \sqrt{dkMT} + T\sqrt{d}\zeta\Big).$$

If each task appears uniformly for $T/M$ rounds, an algorithm that solves each task independently suffers a regret bound of $\widetilde{\Theta}(d\sqrt{MT})$ using a nearly minimax algorithm such as LinUCB (Li et al., 2010). In the case that $k \ll M$ and $k \ll d$, our algorithm saves a factor of $\sqrt{M/k}$ or $\sqrt{M/d}$, which shows it utilizes the underlying representation. Moreover, this regret bound matches the near-optimal algorithm for multi-task linear bandits (Hu et al., 2021), which is a special case of our setting. The following lower bound shows that our algorithm is near-optimal:

**Proposition 5.4** (Lower Bound for Lifelong Linear Bandits with Shared Representation). *Under Assumptions 3.1, 3.2 and 3.6, for any algorithm, there exists a bandit instance for which*

$$\mathbb{E}[\text{Regret}(T)] \geq \Omega\Big(d\sqrt{kT} + k\sqrt{MT} + T\sqrt{d/\log T}\zeta\Big).$$

## 6 CONCLUSION AND FUTURE WORK

In this paper, we proposed a unified framework for linear contextual bandits based on feel-good Thompson sampling, which can cover different variants of linear contextual bandits. At the core of our algorithm is an adaption of feel-good Thompson sampling from reward function to policy class, which enables the algorithm to deal with time-varying environments. We showed that our algorithm can achieve near-optimal regret bounds for these variants of linear bandits, which resolve several open problems in the respective settings.

We notice that there is still a gap between the regret upper and lower bounds in terms of the dependence on the dimension $d$ for non-stationary linear bandits with bounded path length. We leave it as a future work to close this gap.

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

## A  ADDITIONAL RELATED WORK

**Thompson Sampling.** Thompson sampling is a classical algorithm that can be adapted to contextual bandits (Thompson, 1933). Russo & Roy (2014) proposed the first general theoretical guarantee for Thompson sampling. By utilizing the connection between Thompson sampling and optimistic policies, they showed Thompson sampling achieves a Bayesian regret bound of $\widetilde{\mathcal{O}}(d\sqrt{T})$ for linear bandits, where the underlying parameters are randomly drawn from some public distribution. For frequentist regret, Agrawal & Goyal (2013) showed a modified Thompson sampling in which the variance of the posterior distribution is inflated by $\widetilde{\mathcal{O}}(d)$ achieves a regret bound of $\widetilde{\mathcal{O}}(d^{\frac{3}{2}}\sqrt{T})$ on linear bandits. As shown by Hamidi & Bayati (2020), inflation is necessary for Thompson sampling to achieve sub-linear regret; thus, one cannot improve the regret bounds for original Thompson sampling. In comparison, feel-good Thompson sampling (Zhang, 2021) introduced an optimistic bonus on the loss function, making it possible to achieve a nearly minimax regret on the linear contextual bandits.

**Exponentially Weighted Sampling.** A classical technique for online learning is exponentially weighted sampling. This approach allows an algorithm to corral a collection of policies in small regret. By constructing unbiased estimators for unobserved actions, Auer et al. (2002) adapted the ideas to contextual bandits and proposed EXP4. Its regret suffers a polynomial dependence on the number of actions. For linear bandits with stationary contexts, the dependence can be reduced to logarithmic using an EXP2-type algorithm (Bubeck et al., 2012). The algorithm can also be regarded as an EXP3 with a modified estimator, which can be implemented efficiently under specific action sets using stochastic mirror descent.

**UCB-Based Algorithms.** The concept of upper confidence bound was first proposed by Lai & Robbins (1985) to solve multi-arm bandits. The algorithm can be adapted to linear contextual bandits with an infinite number of actions by analyzing the dynamics of the confidence set using an elliptical lemma (Dani et al., 2008; Li et al., 2010; Abbasi-Yadkori et al., 2011). However, it is hard to get the potential lemma for other function spaces, making it difficult to generalize the approach to a broader function class. A recent line of works showed the contextual bandits could be reduced to regression oracles (Foster et al., 2018; Foster & Rakhlin, 2020). This observation pointed out a way for UCB-based algorithms to work on general function classes. However, the algorithm suffers from a polynomial dependence on the number of actions, which is incapable of infinite action sets.

## B  IMPLEMENTATION OF THE PROPOSED ALGORITHM

In this section, we discuss the implementations for our main algorithm FGTS.LP in practice. In particular, we present an efficient algorithm to sample from the distribution defined in (2) approximately. Consider distribution $\widetilde{P}^{(t)}$ in which the probability density function over $\mathbf{\Theta}$ is proportional to

$$\widetilde{P}^{(t)}(\mathbf{\Theta}) \propto \exp\big(-\beta\sum_{\tau=1}^{t-1} L^{(\tau)}(\mathbf{\Theta})\big)\cdot P_0(\mathbf{\Theta}), \tag{5}$$

where $L^{(t)}$ is the loss function defined in (3) and $P_0$ is some prior distribution to be determined. One may see $\widetilde{P}^{(t)}$ is identical to $P^{(t)}(\mathbf{\Theta})$ if one chooses the uniform mixture of discrete distribution of support $\mathcal{H}_\epsilon$ as prior. For efficient implementations, we would use a continuous distribution to cover $\mathcal{H}$ instead. More details can be found in the following sections.

Inspired by Xu et al. (2022), we would use Langevin Monte Carlo (Roberts & Tweedie, 1996; Bakry et al., 2013) to take sample from (5). In specific, on round $t$, we run the following subroutines for $K_t$ steps. For iteration step $s = 1, \cdots, K_t$, we set

$$\mathbf{\Theta}_{t,s-1} = \mathbf{\Theta}_{t,s} - \eta_{t,s}\sum_{\tau=1}^{t-1}\nabla L^{(\tau)}(\mathbf{\Theta}_{t,s-1}) + \beta^{-1}\eta_{t,s}\nabla\log P_0(\mathbf{\Theta}_{t,s-1}) + \sqrt{2\beta^{-1}\eta_{t,s}}\boldsymbol{\epsilon}_{t,s} \tag{6}$$

where $\boldsymbol{\epsilon}_{t,s}$ is Gaussian random matrices and $\eta_{t,s}$ is the step size. This updating rule can be regarded as the Euler-Maruyama discretization of the following Langevin dynamics

$$\mathrm{d}\mathbf{\Theta}(s) = -\sum_{\tau=1}^{t-1}\nabla L^{(\tau)}(\mathbf{\Theta}(s))\mathrm{d}s + \beta^{-1}\nabla\log P_0(\mathbf{\Theta}(s))\mathrm{d}s + \sqrt{2\beta^{-1}}\mathrm{d}\mathbf{B}(s) \tag{7}$$

where $s > 0$ is the continuous time index and $\mathbf{B}(s) \in \mathbb{R}^d$ is a Brownian motion. It was shown under mild conditions, the above Langevin dynamics will converge to the stationary distribution (5). So it is reasonable to take sample from (5) using LMC. We list the corresponding pseudo-code in Algorithm 2.

---
**Algorithm 2** LMC-FGTS.LP, General Framework
---

**input:** prior distribution $P_0$; inverse temperature $\beta$; exploration parameter $\lambda$; number of iterations $\{K_t\}_{t \geq 1}$; step sizes $\{\eta_{t,s}\}_{t,s \geq 1}$.

sample matrix $\boldsymbol{\Theta}^{(0)} \in \mathbb{R}^{d \times T}$ from $\mathcal{H}$ according to $P_0$

**for** round $t = 1, \cdots, T$ **do**

    $\boldsymbol{\Theta}_{t,0} \leftarrow \boldsymbol{\Theta}^{(t-1)}$

    **for** iteration $s = 1, \cdots, K_t$ **do**

        sample matrix $\boldsymbol{\epsilon}_{t,s} \sim \mathcal{N}(0, \mathbf{I}_{T \times d})$

        $\boldsymbol{\Theta}_{t,s} \leftarrow \boldsymbol{\Theta}_{t,s-1} - \eta_{t,s} \sum_{\tau=1}^{t-1} \nabla L^{(\tau)}(\boldsymbol{\Theta}_{t,s-1}) - \beta^{-1} \eta_{t,s} \nabla \log P_0(\boldsymbol{\Theta}_{t,s-1}) + \sqrt{2\beta^{-1}\eta_{t,s}} \boldsymbol{\epsilon}_{t,s}$

    **end for**

    $\boldsymbol{\Theta}^{(t)} \leftarrow \boldsymbol{\Theta}_{t,K_t}$

    take action $\mathbf{a}^{(t)} \leftarrow \arg\max_{\mathbf{a} \in \mathcal{A}^{(t)}} \langle \mathbf{a}, \boldsymbol{\theta}_t^{(t)} \rangle$ where $\boldsymbol{\theta}_t^{(t)}$ is the $t$-th column of $\boldsymbol{\Theta}^{(t)}$

**end for**

---

It is important to mention the above algorithm is a general framework for linear contextual bandits. The algorithm could be further refined into specific settings. For example, since we know linear policy set $\mathcal{H}$ only contains matrix $\boldsymbol{\Theta}$ in which all columns are same, it is sufficient to record only one column of $\boldsymbol{\Theta}$. In the following sections, we discuss how to implement the algorithm for specific linear contextual bandits, and also present corresponding simulation results.

### B.1 MISSPECIFIED LINEAR CONTEXTUAL BANDITS

We first discuss the implementation for misspecified linear contextual bandits, i.e., the variants depicted in Corollary 5.1. In this case, we parameterize any matrix $\boldsymbol{\Theta} = [\boldsymbol{\theta}, \cdots, \boldsymbol{\theta}]$ in $\mathcal{H}$ using any of its column vector $\boldsymbol{\theta}$. We choose the Gaussian distribution $\mathcal{N}(0, \mathbf{I}_d/d)$ as the prior $P_0$. One can check the expected length of random vector sampled from $P_0$ is 1, which matches the hypothesis set given by Assumption 3.1 and 3.3. In the algorithm, we slightly abuse $L^{(t)}$ and use it to denote loss function $\mathbb{R}^d \to \mathbb{R}$,

$$L^{(t)}(\boldsymbol{\theta}) = (r^{(t)} - \langle \mathbf{a}^{(t)}, \boldsymbol{\theta} \rangle)^2 - \lambda \max_{\mathbf{a} \in \mathcal{A}^{(t)}} \langle \mathbf{a}, \boldsymbol{\theta} \rangle. \tag{8}$$

which is a counterpart of (3) for single round policies. We list the pseudo-code in Algorithm 3.

---
**Algorithm 3** LMC-FGTS.LP for Misspecified Linear Contextual Bandits
---

**input:** time horizon $T$, feature dimension $d$, misspecification level $\xi$, inverse temperature $\beta$; exploration parameter $\lambda$; number of iterations $\{K_t\}_{t \geq 1}$; step sizes $\{\eta_{t,s}\}_{t,s \geq 1}$.

sample vector $\boldsymbol{\theta}^{(0)} \sim \mathcal{N}(0, \mathbf{I}_d/d)$

**for** round $t = 1, \cdots, T$ **do**

    $\boldsymbol{\theta}_{t,0} \leftarrow \boldsymbol{\theta}^{(t-1)}$

    **for** iteration $s = 1, \cdots, K_t$ **do**

        sample vector $\boldsymbol{\epsilon}_{t,s} \sim \mathcal{N}(0, \mathbf{I}_d)$

        $\boldsymbol{\theta}_{t,s} \leftarrow \boldsymbol{\theta}_{t,s-1} - \eta_{t,s} \sum_{\tau=1}^{t-1} \nabla L^{(\tau)}(\boldsymbol{\theta}_{t,s-1}) - d\beta^{-1}\eta_{t,s}\boldsymbol{\theta}_{t,s-1} + \sqrt{2\beta^{-1}\eta_{t,s}} \boldsymbol{\epsilon}_{t,s}$

    **end for**

    $\boldsymbol{\theta}^{(t)} \leftarrow \boldsymbol{\theta}_{t,K_t}$

    take action $\mathbf{a}^{(t)} \leftarrow \arg\max_{\mathbf{a} \in \mathcal{A}^{(t)}} \langle \mathbf{a}, \boldsymbol{\theta}^{(t)} \rangle$

**end for**

---

**Simulation Results.** We generate the synthetic data in the following way: The horizon length is set to $T = 3000$ and the feature dimension is set to $d = 100$. We first generate $\boldsymbol{\theta}_0$ from unit sphere $S^{d-1} = \{\mathbf{x} \in \mathbb{R}^d : \|\mathbf{x}\|_2 = 1\}$ randomly. On each round $t$, $A = 20$ actions $\{\mathbf{a}_i^{(t)}\}_{i=1\ldots A}$ are independently drawn from the same unit sphere. The reward of action $\mathbf{a}_i^{(t)}$ is set to $f^{(t)}(\mathbf{a}^{(t)}) = \langle \mathbf{a}_i^{(t)}, \boldsymbol{\theta}_0 \rangle + \text{Unif}(-\zeta, \zeta)$ where $\text{Unif}(l, u)$ is the uniform random distribution in $[l, u]$ for misspecification level $\zeta \in \{0, 0.05, 0.1\}$. The observed reward of it is then given by $r^{(t)} = f^{(t)}(\mathbf{a}^{(t)}) + \xi^{(t)}$ where $\xi^{(t)}$ is randomly sampled from Gaussian noise with standard deviation 0.2.

We run Algorithm 3 with exploration parameter $\lambda = \sqrt{1/T + \zeta^2/d}$ according to our theoretical result given by Theorem 5.1 and Lemma 5.1. We set the temperature parameter $\beta^{-1}$ to 0.01 following Xu et al. (2022), which is the $1/4$ of the variance of reward noise. On each round $t$, the algorithm do $K_t = 200$ iterations and the step size of iteration $s$ is $\tau_{t,s} = 1/(ds)$. To accelerate the running speed of the algorithm, we do not find the precise maximum action in (8) in each iteration. We record the maximum action and recompute it only in first 14 steps or once for every 14 steps afterwards.

We compare our algorithm to LinUCB with adaptation to misspecification (Zanette et al., 2020) and Linear Thompson Sampling (Abeille & Lazaric, 2017). The result is shown in Figure 1. One can see our algorithm has comparative performance with those existing algorithms.

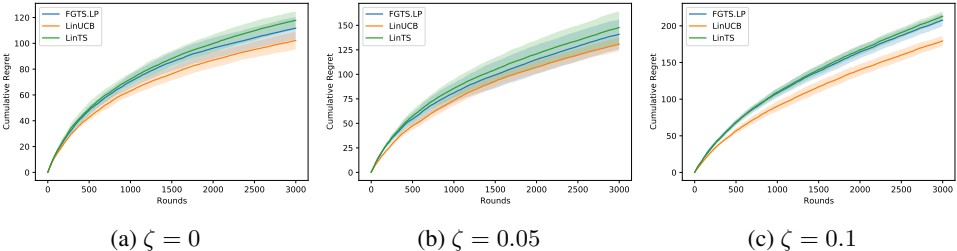

(a) $\zeta = 0$  (b) $\zeta = 0.05$  (c) $\zeta = 0.1$

Figure 1: The performance of Algorithm 3 comparing with other linear contextual bandit algorithms. Results are averaged over 5 runs with standard errors shown as shaded areas.

## B.2  Lifelong Linear Contextual Bandits

This section discuss the practical implementation for lifelong linear bandits, i.e., the variants depicted in Corollary 5.4. In this case, the linear policy set $\mathcal{H}$ only contains matrices $\Theta$ that can be decomposed using two matrices $\mathbf{B} \in \mathbb{R}^{d \times k}$ and $\mathbf{W} = \{\mathbf{w}_i\}_{i=1}^M \in \mathbb{R}^{M \times k}$. Thus, we do not need to maintain the whole sequence of models $\Theta$, it is sufficient to keep those two matrices. Moreover, rather than taking LMC iterations on $\Theta$, we compute the gradient respective to $\mathbf{B}$ and $\mathbf{W}$ to update them directly.

In the algorithm, instead of taking LMC iterations on $\Theta$, we compute the gradient to update $\mathbf{B}$ and $\mathbf{W}$ directly. The prior distribution is also defined on those two matrices. We take $\mathcal{N}(0, k\mathbf{I}_{d \times k}/d)$ and $\mathcal{N}(0, \mathbf{I}_{k \times M}/k)$ as the prior distribution for $\mathbf{B}$ and $\mathbf{W}$, respectively. We use a different prior distribution for initialization which ensures the length of $\theta$ is about 1 to improve the overall performance. On every iteration of round $t$, the algorithm computes linear policy for all historical task $\theta_{t,s-1,i}$ and uses them to compute gradient. The loss function is defined in (8). We list the pseudo-code in Algorithm 4.

---

**Algorithm 4** LMC-FGTS.LP for (Generalized) Lifelong Linear Contextual Bandits

---

**input:** task index sequence $\{m^{(t)}\}_{t=1}^T$; inverse temperature $\beta$; exploration parameter $\lambda$; number of iterations $\{K_t\}_{t \geq 1}$; step sizes $\{\eta_{t,s}\}_{t,s \geq 1}$.

sample matrix $\mathbf{B}^{(0)}$ from orthogonal random matrices in $\mathbb{R}^{d \times k}$

sample matrix $\mathbf{W}^{(0)} \sim \mathcal{N}(0, \mathbf{I}_{k \times M}/k)$

**for** round $t = 1, \cdots, T$ **do**

  $(\mathbf{B}_{t,0}, \mathbf{W}_{t,0}) \leftarrow (\mathbf{B}^{(t-1)}, \mathbf{W}^{(t-1)})$

  **for** iteration $s = 1, \cdots, K_t$ **do**

    sample matrix $\boldsymbol{\epsilon}_{t,s}^B \sim \mathcal{N}(0, \mathbf{I}_{d \times k})$ and $\boldsymbol{\epsilon}_{t,s}^W \sim \mathcal{N}(0, \mathbf{I}_{k \times M})$

    compute $\theta_{t,s-1,i} \leftarrow \mathbf{B}_{t,s-1}, \mathbf{w}_{t,s-1,i}$ for $i \in [M]$ where $\mathbf{w}_{t,s-1,i}$ is the $i$-th row of $\mathbf{W}_{t,s-1}$

    $\mathbf{B}_{t,s} \leftarrow \mathbf{B}_{t,s-1} - \eta_{t,s} \sum_{\tau=1}^{t-1} \nabla_{\mathbf{B}} L^{(\tau)}(\theta_{t,s,m^{(\tau)}}) - dk^{-1}\beta^{-1}\eta_{t,s}\mathbf{B}_{t,s-1} + \sqrt{2\beta^{-1}\eta_{t,s}}\boldsymbol{\epsilon}_{t,s}^B$

    $\mathbf{W}_{t,s} \leftarrow \mathbf{W}_{t,s-1} - \eta_{t,s} \sum_{\tau=1}^{t-1} \nabla_{\mathbf{W}} L^{(\tau)}(\theta_{t,s,m^{(\tau)}}) - k\beta^{-1}\eta_{t,s}\mathbf{W}_{t,s-1} + \sqrt{2\beta^{-1}\eta_{t,s}}\boldsymbol{\epsilon}_{t,s}^W$

  **end for**

  $(\mathbf{B}^{(t)}, \mathbf{W}^{(t)}) \leftarrow (\mathbf{B}_{t,K_t}, \mathbf{W}_{t,K_t})$

  compute $\theta^{(t)} \leftarrow \mathbf{B}^{(t)}\mathbf{w}_{m^{(t)}}^{(t)}$ where $\mathbf{w}_{m^{(t)}}^{(t)}$ is the $m^{(t)}$-th row of $\mathbf{W}^{(t)}$

  take action $\mathbf{a}^{(t)} \leftarrow \arg\max_{\mathbf{a} \in \mathcal{A}^{(t)}} \langle \mathbf{a}, \theta^{(t)} \rangle$

**end for**

---

**Simulation Results.**  We generate the synthetic data in the following way: The horizon length is set to $T = 3000$ and the feature dimension is set to $d = 40$. There are $M = 20$ tasks and each of them appear in exactly $T/M = 100$ rounds in sequential, i.e, the task on round $t$ is given by

$m^{(t)} = \lceil t/100 \rceil$. The dimension of hidden subspace is set in $k \in \{2, 3, 4\}$. The hidden linear feature extractor $\mathbf{B}$ is a random orthogonal matrix in $\mathbb{R}^{d \times k}$ and each of the task-specific vector $\{\mathbf{w}_i\}_{i=1}^M$ are independent random vectors from unit sphere $S^{k-1} = \{\mathbf{x} \in \mathbb{R}^k : \|\mathbf{x}\|_2 = 1\}$. On each round $t$, $A = 20$ actions $\{\mathbf{a}_i^{(t)}\}_{i=1\cdots A}$ are randomly drawn from unit sphere $S^{d-1}$. We fix the misspecification level to be $\zeta = 0$. The observed reward of action $\mathbf{a}_i^{(t)}$ is given by $r^{(t)} = \langle \mathbf{a}_i^{(t)}, \mathbf{B}\mathbf{w}_{m^{(t)}} \rangle + \xi^{(t)}$ where $\xi^{(t)}$ is independent random Gaussian variable of standard deviation 0.2. We run Algorithm 4 with exploration parameter $\lambda = \sqrt{(dk + Mk)/(dT)}$ and temperature parameter $\beta^{-1} = 0.01$. On each round $t$, the algorithm do $K_t = 100$ iterations and the step size of iteration $s$ is $\tau_{t,s} = 1/(dks)$. Similar to misspecified linear contextual bandits, we record the maximum action and recompute it only in first 10 steps or once for every 10 steps afterwards to accelerate its running speed. Since there is no existing algorithm for lifelong linear contextual bandits, we compare the performance with running LinUCB on each task independently. The result is shown in figure 2. One can see our algorithm leverage the underlying representation with a good constant factor.

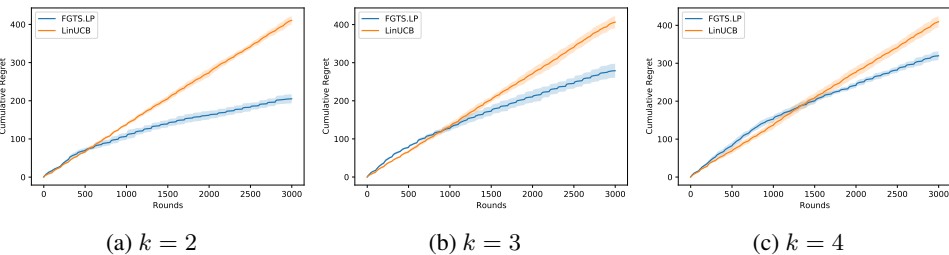

(a) $k = 2$          (b) $k = 3$          (c) $k = 4$

Figure 2: The performance of Algorithm 4 comparing with running LinUCB on each task. Results are averaged over 5 runs with standard errors shown as shaded areas.

## C    PROOF SKETCH OF THEOREM 5.1

This section presents a proof sketch of our main Theorem 5.1. We note the proof is similar to the proof of Theorem 2 in Zhang (2021). The main difference comes from the usage of covering number of the policy set and our decoupling lemma under misspecification.

Let $\mathbf{\Theta}^* = [\boldsymbol{\theta}_1^*, \cdots, \boldsymbol{\theta}_T^*]$ be the optimal policy in the $\epsilon$-net $\mathcal{H}_\epsilon$. For any linear policy $\mathbf{\Theta} = [\boldsymbol{\theta}_1, \cdots, \boldsymbol{\theta}_T]$, denote $\Delta_t(\mathbf{\Theta}, \mathbf{a}^{(t)}) := \langle \mathbf{a}^{(t)}, \boldsymbol{\theta}_t - \boldsymbol{\theta}_t^* \rangle$ as the Bellman error and $\Delta_t^*(\mathbf{\Theta}) := \max_{\mathbf{a} \in \mathcal{A}^{(t)}} \langle \mathbf{a}, \boldsymbol{\theta}_t \rangle - \max_{\mathbf{a} \in \mathcal{A}^{(t)}} \langle \mathbf{a}, \boldsymbol{\theta}_t^* \rangle$ as the feel-good error. Since $\langle \mathbf{a}^{(t)}, \boldsymbol{\theta}_t \rangle = \max_{\mathbf{a} \in \mathcal{A}^{(t)}} \langle \mathbf{a}, \boldsymbol{\theta}_t \rangle$ according to definition of $\mathbf{a}^{(t)}$, we can decompose the expected regret on round $t$ by

$$\mathbb{E}_{\mathbf{a}^{(t)}} [f^{(t)}(\mathbf{a}_*^{(t)}) - f^{(t)}(\mathbf{a}^{(t)})] \leq \mathbb{E}_{\mathbf{a}^{(t)}, \mathbf{\Theta}^{(t)}} \left[ \Delta_t(\mathbf{\Theta}^{(t)} \mathbf{a}^{(t)}) - \Delta_t^*(\mathbf{\Theta}^{(t)}) \right] + 2(\epsilon + \zeta). \tag{9}$$

Besides, we construct a potential function following the classical analysis for exponentially weighted sampling,

$$\Phi^{(t)} = \frac{1}{\beta\lambda} \log \sum_{\mathbf{\Theta} \in \mathcal{H}_\epsilon} \exp\left( \beta \sum_{\tau=1}^t \left( L^{(\tau)}(\mathbf{\Theta}^*) - L^{(\tau)}(\mathbf{\Theta}) \right) \right).$$

The boundary value of the potential function follows

$$\Phi^{(0)} - \Phi^{(T)} \leq \frac{\log |\mathcal{H}_\epsilon|}{\beta\lambda}. \tag{10}$$

The increment of potential function on round $t$ satisfies

$$\mathbb{E}_{\mathbf{a}^{(t)}, r^{(t)}} \left[ \Phi^{(t)} - \Phi^{(t-1)} \right] \leq \frac{1}{\beta\lambda} \mathbb{E}_{\mathbf{a}^{(t)}} \left[ \log \mathbb{E}_{r^{(t)}} \mathbb{E}_{\mathbf{\Theta}} \exp\left( \beta\left( L^{(t)}(\mathbf{\Theta}^*) - L^{(t)}(\mathbf{\Theta}) \right) \right) \right]. \tag{11}$$

Recall $L^{(t)}(\mathbf{\Theta}) = (r^{(t)} - \langle \mathbf{a}^{(t)}, \boldsymbol{\theta}_t \rangle)^2 - \lambda \max_{\mathbf{a} \in \mathcal{A}^{(t)}} \langle \mathbf{a}, \boldsymbol{\theta}_t \rangle$, the randomness on $r^{(t)}$ only casts an additive noise $-2\beta\xi^{(t)}\Delta_t(\mathbf{\Theta}, \mathbf{a}^{(t)})$ to the term $\beta\left( L^{(t)}(\mathbf{\Theta}^*) - L^{(t)}(\mathbf{\Theta}) \right)$ in the exponent. Since $\xi^{(t)}$ is constantly bounded and zero mean, the noise can be controlled by selecting proper $\beta$ according

to Hoeffding's lemma. With classical inequalities over logarithm and exponential, we can show the variation of potential function satisfies

$$\mathbb{E}_{\mathbf{a}^{(t)}, r^{(t)}} \left[ \Phi^{(t)} - \Phi^{(t-1)} \right] \leq -\frac{1}{\lambda} \mathbb{E}_{\mathbf{a}^{(t)}, \boldsymbol{\Theta}} \left[ \Delta_t(\boldsymbol{\Theta}, \mathbf{a}^{(t)})^2 + C_t(\mathbf{a}^{(t)}) \Delta_t(\boldsymbol{\Theta}, \mathbf{a}^{(t)}) \right]$$
$$+ \mathbb{E}_{\boldsymbol{\Theta}} \Delta_t^*(\boldsymbol{\Theta}) + O\left( \beta\lambda + \frac{\beta(\epsilon + \zeta)^2}{\lambda} \right) \tag{12}$$

where $C_t(\mathbf{a}^{(t)}) = f^{(t)} - \langle \boldsymbol{\Theta}^{(t)}, \boldsymbol{\theta}_t^* \rangle$ is the misspecification of $\boldsymbol{\Theta}^*$ with respect to ground truth $f^{(t)}$. One can see that the first two terms on the RHS of (12) are very similar to the first two terms on the RHS of (9). But with one critical difference, the chosen policy $\boldsymbol{\Theta}^{(t)}$ in (9) is replaced by a random policy $\boldsymbol{\Theta}$ in (12): Although $\boldsymbol{\Theta}^{(t)}$ and $\boldsymbol{\Theta}$ have the same margin, $\boldsymbol{\Theta}^{(t)}$ is correlated with $\mathbf{a}^{(t)}$ but $\boldsymbol{\Theta}$ is independent from $\mathbf{a}^{(t)}$. As a result, $\boldsymbol{\Theta}^{(t)} \Delta_t^*(\boldsymbol{\Theta}^{(t)}) = \mathbb{E}_{\boldsymbol{\Theta}} \Delta_t^*(\boldsymbol{\Theta})$ but $\mathbb{E}_{\mathbf{a}^{(t)}, \boldsymbol{\Theta}^{(t)}} [\Delta_t(\boldsymbol{\Theta}^{(t)}, \mathbf{a}^{(t)})]$ is not necessary equal to $\mathbb{E}_{\mathbf{a}^{(t)}, \boldsymbol{\Theta}} [\Delta_t(\boldsymbol{\Theta}, \mathbf{a}^{(t)})]$. Therefore, we cannot get a regret bound by combining (9), (10) and (12) directly.

This is where the decoupling lemma helps. Following Zhang (2021), one can get the following lemma for linear models,

$$\mathbb{E}_{\mathbf{a}^{(t)}, \boldsymbol{\Theta}^{(t)}} \Delta_t(\boldsymbol{\Theta}^{(t)}, \mathbf{a}^{(t)}) \leq \frac{1}{\lambda} \mathbb{E}_{\mathbf{a}^{(t)}, \boldsymbol{\Theta}} \Delta_t(\boldsymbol{\Theta}, \mathbf{a}^{(t)})^2 + \mathcal{O}(\lambda d). \tag{13}$$

For our usage, in order to achieve optimal dependence with the covering radius and misspecification, we derived a decoupling lemma includes the corruptions. In particular, Lemma D.2 implies

$$\mathbb{E}_{\mathbf{a}^{(t)}, \boldsymbol{\Theta}^{(t)}} \Delta_t(\boldsymbol{\Theta}^{(t)}, \mathbf{a}^{(t)}) \leq \frac{1}{\lambda} \mathbb{E}_{\mathbf{a}^{(t)}, \boldsymbol{\Theta}} \left[ \Delta_t(\boldsymbol{\Theta}, \mathbf{a}^{(t)})^2 + C_t(\mathbf{a}^{(t)}) \Delta_t(\boldsymbol{\Theta}, \mathbf{a}^{(t)}) \right]$$
$$+ O\left( \lambda d + (\epsilon + \zeta) + \frac{(\epsilon + \zeta)^2}{\lambda} \right). \tag{14}$$

By combining (9), (10), (12) and (14), with $\beta = \Theta(1)$, we conclude that

$$\mathbb{E}[\text{Regret}^{(T)}] = \sum_{t=1}^T \mathbb{E}_{\mathbf{a}^{(t)}} [f^{(t)}(\mathbf{a}_*^{(t)}) - f^{(t)}(\mathbf{a}^{(t)})] \leq O\left( \lambda d T + (\epsilon + \zeta)T + \frac{\log|\mathcal{H}_\epsilon| + (\epsilon + \zeta)^2 T}{\lambda} \right).$$

One can obtain the desired statement using $\lambda = \Theta\left( \sqrt{\log|\mathcal{H}_\epsilon|/(dT) + (\epsilon + \zeta)^2/d} \right)$.

We note our analysis uses a refined way to deal with the covering radius and misspecification. One can only get a regret bound of $\mathcal{O}(\sqrt{dT \log|\mathcal{H}_\epsilon|} + T\sqrt{d(\epsilon + \zeta)})$ if uses (13) and follows the proof in Zhang (2021) faithfully, which has a worse dependence on $(\epsilon + \zeta)$.

## D FULL PROOF OF THEOREM 5.1

In this section, we will provide a complete proof of Theorem 5.1. The following decoupling lemma is a critical structural lemma for our analysis, which is a generalization of Lemma 2 in Zhang (2021).

**Lemma D.1** (Decoupling Lemma). *Let $P$ be a joint distribution over two $\mathbb{R}^d$ space, i.e., $P \in \Delta(\mathbb{R}^d \times \mathbb{R}^d)$. For any constant $\lambda > 0$, we have*

$$\mathbb{E}_{(\boldsymbol{\theta}, \boldsymbol{\phi}) \sim P} \langle \boldsymbol{\theta}, \boldsymbol{\phi} \rangle \leq d\lambda + \frac{0.25}{\lambda} \mathbb{E}_{\substack{(\boldsymbol{\theta}, \boldsymbol{\phi}) \sim P \\ (\boldsymbol{\theta}', \boldsymbol{\phi}') \sim P}} \langle \boldsymbol{\theta}, \boldsymbol{\phi}' \rangle^2,$$

*where $(\boldsymbol{\theta}, \boldsymbol{\phi})$ on LHS is a sample from $P$ while $(\boldsymbol{\theta}, \boldsymbol{\phi})$ and $(\boldsymbol{\theta}', \boldsymbol{\phi}')$ on RHS are two independent samples from $P$.*

*Proof.* Let $\boldsymbol{\Sigma} = \mathbb{E}_{(\boldsymbol{\theta},\boldsymbol{\phi})\sim P}\,\boldsymbol{\theta}\boldsymbol{\theta}^\top$ be a matrix in $\mathbb{R}^{d\times d}$. Denote $\boldsymbol{\xi}_1,\cdots,\boldsymbol{\xi}_d$ as a set of orthogonal eigenvectors of $\boldsymbol{\Sigma}$. Let $s_i = \mathbb{E}_{(\boldsymbol{\theta},\boldsymbol{\phi})\sim P}\langle\boldsymbol{\theta},\boldsymbol{\xi}_i\rangle^2$. Then,

$$
\begin{aligned}
\mathbb{E}_{(\boldsymbol{\theta},\boldsymbol{\phi})\sim P}\langle\boldsymbol{\theta},\boldsymbol{\phi}\rangle &= \mathbb{E}_{(\boldsymbol{\theta},\boldsymbol{\phi})\sim P}\sum_{i=1}^d\langle\boldsymbol{\theta},\boldsymbol{\xi}_i\rangle\langle\boldsymbol{\xi}_i,\boldsymbol{\phi}\rangle \\
&= \sum_{i=1}^d \mathbb{E}_{(\boldsymbol{\theta},\boldsymbol{\phi})\sim P}\sqrt{\frac{2\lambda}{s_i}}\langle\boldsymbol{\theta},\boldsymbol{\xi}_i\rangle\cdot\sqrt{\frac{0.5s_i}{\lambda}}\langle\boldsymbol{\xi}_i,\boldsymbol{\phi}\rangle \\
&\leq \sum_{i=1}^d \mathbb{E}_{(\boldsymbol{\theta},\boldsymbol{\phi})\sim P}\Big(\frac{\lambda}{s_i}\langle\boldsymbol{\theta},\boldsymbol{\xi}_i\rangle^2 + \frac{0.25s_i}{\lambda}\langle\boldsymbol{\xi}_i,\boldsymbol{\phi}\rangle^2\Big),
\end{aligned}
\tag{15}
$$

where the first equality follows from $\boldsymbol{\theta} = \sum_{i=1}^d\langle\boldsymbol{\theta},\boldsymbol{\xi}_i\rangle\boldsymbol{\xi}_i$ and the inequality follows from $ab \leq 0.5a^2 + 0.5b^2$. For the first term, we have

$$
\sum_{i=1}^d \mathbb{E}_{(\boldsymbol{\theta},\boldsymbol{\phi})\sim P}\frac{\lambda}{s_i}\langle\boldsymbol{\theta},\boldsymbol{\xi}_i\rangle^2 = \sum_{i=1}^d\lambda = \lambda d.
\tag{16}
$$

For the second term, it holds that

$$
\begin{aligned}
\frac{0.25}{\lambda}\sum_{i=1}^d \mathbb{E}_{(\boldsymbol{\theta},\boldsymbol{\phi})\sim P}s_i\langle\boldsymbol{\xi}_i,\boldsymbol{\phi}\rangle^2 &= \frac{0.25}{\lambda}\sum_{i=1}^d \mathbb{E}_{\substack{(\boldsymbol{\theta},\boldsymbol{\phi})\sim P \\ (\boldsymbol{\theta}',\boldsymbol{\phi}')\sim P}}\langle\boldsymbol{\theta},\boldsymbol{\xi}_i\rangle^2\langle\boldsymbol{\xi}_i,\boldsymbol{\phi}'\rangle^2 \\
&= \frac{0.25}{\lambda}\sum_{i=1}^d\sum_{j=1}^d \mathbb{E}_{\substack{(\boldsymbol{\theta},\boldsymbol{\phi})\sim P \\ (\boldsymbol{\theta}',\boldsymbol{\phi}')\sim P}}\langle\boldsymbol{\theta},\boldsymbol{\xi}_i\rangle\langle\boldsymbol{\xi}_i,\boldsymbol{\phi}'\rangle\langle\boldsymbol{\theta},\boldsymbol{\xi}_j\rangle\langle\boldsymbol{\xi}_j,\boldsymbol{\phi}'\rangle \\
&= \frac{0.25}{\lambda} \mathbb{E}_{\substack{(\boldsymbol{\theta},\boldsymbol{\phi})\sim P \\ (\boldsymbol{\theta}',\boldsymbol{\phi}')\sim P}}\Big(\sum_{i=1}^d\langle\boldsymbol{\theta},\boldsymbol{\xi}_i\rangle\langle\boldsymbol{\xi}_i,\boldsymbol{\phi}'\rangle\Big)^2 \\
&= \frac{0.25}{\lambda} \mathbb{E}_{\substack{(\boldsymbol{\theta},\boldsymbol{\phi})\sim P \\ (\boldsymbol{\theta}',\boldsymbol{\phi}')\sim P}}\langle\boldsymbol{\theta},\boldsymbol{\phi}'\rangle^2,
\end{aligned}
\tag{17}
$$

where the first equality follows from the definition of $s_i$, the second equality holds since $\mathbb{E}_{(\boldsymbol{\theta},\boldsymbol{\phi})\sim P}\langle\boldsymbol{\theta},\boldsymbol{\xi}_i\rangle\langle\boldsymbol{\theta},\boldsymbol{\xi}_j\rangle = \boldsymbol{\xi}_i^\top\boldsymbol{\Sigma}\boldsymbol{\xi}_j = 0$ for all $i \neq j$ and the final equality follows from the definition of $\boldsymbol{\xi}_i$. By plugging (16) and (17) into (15), we can conclude that

$$
\mathbb{E}_{(\boldsymbol{\theta},\boldsymbol{\phi})\sim P}\langle\boldsymbol{\theta},\boldsymbol{\phi}\rangle \leq \lambda d + \frac{0.25}{\lambda} \mathbb{E}_{\substack{(\boldsymbol{\theta},\boldsymbol{\phi})\sim P \\ (\boldsymbol{\theta}',\boldsymbol{\phi}')\sim P}}\langle\boldsymbol{\theta},\boldsymbol{\phi}'\rangle^2.
$$

$\square$

Using this lemma, we can bound the expectation of the inner product of two correlated random variables using the expectation of the inner product of two independent random variables with the corresponding margin. To provide a nearly minimax bound with optimal dependence on misspecification, we further propose a decoupling lemma that takes misspecification into account:

**Lemma D.2** (Decoupling Lemma with Misspecification). *Let $P$ be a joint distribution over two $\mathbb{R}^d$ space, i.e., $P \in \Delta(\mathbb{R}^d\times\mathbb{R}^d)$. Let $C(\cdot) : \mathbb{R}^d \to \mathbb{R}$ be a function such that $|C(\boldsymbol{\theta})| \leq \zeta$ for all $\boldsymbol{\theta} \in \mathbb{R}^d$. For any constant $\lambda > 0$, we have*

$$
\mathbb{E}_{(\boldsymbol{\theta},\boldsymbol{\phi})\sim P}\langle\boldsymbol{\theta},\boldsymbol{\phi}\rangle \leq \lambda(d+1) + \frac{4\zeta^2}{\lambda} + 4\zeta + \frac{0.25}{\lambda} \mathbb{E}_{\substack{(\boldsymbol{\theta},\boldsymbol{\phi})\sim P \\ (\boldsymbol{\theta}',\boldsymbol{\phi}')\sim P}}\Big[\langle\boldsymbol{\theta},\boldsymbol{\phi}'\rangle^2 + 8C(\boldsymbol{\theta})\cdot\langle\boldsymbol{\theta},\boldsymbol{\phi}'\rangle\Big],
$$

*where $(\boldsymbol{\theta},\boldsymbol{\phi})$ on LHS is a sample from $P$ while $(\boldsymbol{\theta},\boldsymbol{\phi})$ and $(\boldsymbol{\theta}',\boldsymbol{\phi}')$ on RHS are two independent samples from $P$.*

*Proof.* Let $P_+$ be an auxiliary distribution over $\Delta(\mathbb{R}^{d+1}, \mathbb{R}^{d+1})$ in which each element is $(\boldsymbol{\theta}_+, \boldsymbol{\phi}_+) = ([\boldsymbol{\theta}^\top, C(\boldsymbol{\theta})]^\top, [\boldsymbol{\phi}^\top, 1]^\top)$ where $(\boldsymbol{\theta}, \boldsymbol{\phi}) \sim P$. For two independent samples $(\boldsymbol{\theta}_+, \boldsymbol{\phi}_+)$ and $(\boldsymbol{\theta}'_+, \boldsymbol{\phi}'_+)$, we have

$$\langle \boldsymbol{\theta}_+, \boldsymbol{\phi}_+ \rangle = \langle \boldsymbol{\theta}, \boldsymbol{\phi} \rangle + 4C(\boldsymbol{\theta}), \tag{18}$$

$$\langle \boldsymbol{\theta}_+, \boldsymbol{\phi}'_+ \rangle^2 = \langle \boldsymbol{\theta}, \boldsymbol{\phi}' \rangle^2 + 8C(\boldsymbol{\theta}) \cdot \langle \boldsymbol{\theta}, \boldsymbol{\phi}' \rangle + 16C(\boldsymbol{\theta})^2. \tag{19}$$

Apply Lemma D.1 to distribution $P_+$, we have

$$\mathop{\mathbb{E}}_{(\boldsymbol{\theta}_+, \boldsymbol{\phi}_+) \sim P_+} \langle \boldsymbol{\theta}_+, \boldsymbol{\phi}_+ \rangle \leq \lambda(d+1) + \frac{0.25}{\lambda} \mathop{\mathbb{E}}_{\substack{(\boldsymbol{\theta}_+, \boldsymbol{\phi}_+) \sim P_+ \\ (\boldsymbol{\theta}'_+, \boldsymbol{\phi}'_+) \sim P_+}} \langle \boldsymbol{\theta}_+, \boldsymbol{\phi}'_+ \rangle^2. \tag{20}$$

Plugging (18) and (19) into (20), we have

$$\mathop{\mathbb{E}}_{(\boldsymbol{\theta}, \boldsymbol{\phi}) \sim P} \left[ \langle \boldsymbol{\theta}, \boldsymbol{\phi} \rangle + 4C(\boldsymbol{\theta}) \right] \leq \lambda(d+1) + \frac{0.25}{\lambda} \mathop{\mathbb{E}}_{\substack{(\boldsymbol{\theta}, \boldsymbol{\phi}) \sim P \\ (\boldsymbol{\theta}', \boldsymbol{\phi}') \sim P}} \left[ \langle \boldsymbol{\theta}, \boldsymbol{\phi}' \rangle^2 + 8C(\boldsymbol{\theta}) \cdot \langle \boldsymbol{\theta}, \boldsymbol{\phi}' \rangle + 16C(\boldsymbol{\theta})^2 \right].$$

Since $|C(\boldsymbol{\theta})| \leq \zeta$ always holds, we can further conclude that

$$\mathop{\mathbb{E}}_{(\boldsymbol{\theta}, \boldsymbol{\phi}) \sim P} \langle \boldsymbol{\theta}, \boldsymbol{\phi} \rangle \leq \lambda(d+1) + \frac{4\zeta^2}{\lambda} + 4\zeta + \frac{0.25}{\lambda} \mathop{\mathbb{E}}_{\substack{(\boldsymbol{\theta}, \boldsymbol{\phi}) \sim P \\ (\boldsymbol{\theta}', \boldsymbol{\phi}') \sim P}} \left[ \langle \boldsymbol{\theta}, \boldsymbol{\phi}' \rangle^2 + 8C(\boldsymbol{\theta}) \cdot \langle \boldsymbol{\theta}, \boldsymbol{\phi}' \rangle \right].$$

$\square$

Denote $\Omega_-^{(t)} = \Omega^{(t-1)} \cup \{\mathcal{A}^{(t)}, f^{(t)}\}$ be the history before the agent chooses action on round $t$. Let $\boldsymbol{\Theta}^* = [\boldsymbol{\theta}_1^*, \cdots, \boldsymbol{\theta}_T^*]$ be the optimal policy within $\epsilon$-net $\mathcal{H}_\epsilon$. For any linear policy $\boldsymbol{\Theta} = [\boldsymbol{\theta}_1, \cdots, \boldsymbol{\theta}_T]$, denote

$$\Delta_t(\boldsymbol{\Theta}, \mathbf{a}^{(t)}) := \langle \mathbf{a}^{(t)}, \boldsymbol{\theta}_t - \boldsymbol{\theta}_t^* \rangle, \Delta_t^*(\boldsymbol{\Theta}) := \max_{\mathbf{a} \in \mathcal{A}^{(t)}} \langle \mathbf{a}, \boldsymbol{\theta}_t \rangle - \max_{\mathbf{a} \in \mathcal{A}^{(t)}} \langle \mathbf{a}, \boldsymbol{\theta}_t^* \rangle.$$

Note $\Delta_t$ is referred to the Bellman error and $\Delta_t^*$ is referred to the feel-good error. The next lemma shows that we can decomposed the expected regret using these two notions:

**Lemma D.3.** *Under Assumption 3.1, the expected regret on round $t$ satisfies*

$$\mathop{\mathbb{E}}_{\mathbf{a}^{(t)} | \Omega_-^{(t)}} \left[ f^{(t)}(\mathbf{a}_*^{(t)}) - f^{(t)}(\mathbf{a}^{(t)}) \right] \leq \mathop{\mathbb{E}}_{\mathbf{a}^{(t)}, \boldsymbol{\Theta}^{(t)} | \Omega_-^{(t)}} \left[ \Delta_t(\boldsymbol{\Theta}^{(t)}, \mathbf{a}^{(t)}) - \Delta_t^*(\boldsymbol{\Theta}^{(t)}) \right] + 2(\epsilon + \zeta).$$

*Proof.* For any action $\mathbf{a} \in \mathcal{A}$, we have

$$\langle \mathbf{a}, \boldsymbol{\theta}_t^* \rangle \leq f^{(t)}(\mathbf{a}) + |f^{(t)}(\mathbf{a}) - \langle \mathbf{a}, \boldsymbol{\theta}(f^{(t)}) \rangle| + |\langle \mathbf{a}, \boldsymbol{\theta}(f^{(t)}) \rangle - \langle \mathbf{a}, \boldsymbol{\theta}_t^* \rangle| \leq f^{(t)}(\mathbf{a}) + \epsilon + \zeta,$$

where the first inequality follows from triangle inequality and the second inequality holds follows from Assumption 3.1 and the definition of $\epsilon$-net with $|\langle \mathbf{a}, \boldsymbol{\theta}(f^{(t)}) \rangle - \langle \mathbf{a}, \boldsymbol{\theta}_t^* \rangle| \leq \|\mathbf{a}\|_2 \cdot \|\boldsymbol{\theta}(f^{(t)}) - \boldsymbol{\theta}_t^*\|_2 \leq \zeta$. Similarly, we have $\langle \mathbf{a}, \boldsymbol{\theta}_t^* \rangle \geq f^{(t)}(\mathbf{a}) - (\epsilon + \zeta)$. Thus,

$$\mathop{\mathbb{E}}_{\mathbf{a}^{(t)} | \Omega_-^{(t)}} \left[ f^{(t)}(\mathbf{a}_*^{(t)}) - f^{(t)}(\mathbf{a}^{(t)}) \right] \leq \mathop{\mathbb{E}}_{\mathbf{a}^{(t)} | \Omega_-^{(t)}} \left[ \langle \mathbf{a}_*^{(t)}, \boldsymbol{\theta}_t^* \rangle - \langle \mathbf{a}^{(t)}, \boldsymbol{\theta}_t^* \rangle \right] + 2(\epsilon + \zeta). \tag{21}$$

Moreover, we have

$$\mathop{\mathbb{E}}_{\mathbf{a}^{(t)} | \Omega_-^{(t)}} \left[ \langle \mathbf{a}_*^{(t)}, \boldsymbol{\theta}_t^* \rangle - \langle \mathbf{a}^{(t)}, \boldsymbol{\theta}_t^* \rangle \right]$$

$$\leq \mathop{\mathbb{E}}_{\mathbf{a}^{(t)} | \Omega_-^{(t)}} \left[ \max_{\mathbf{a} \in \mathcal{A}^{(t)}} \langle \mathbf{a}, \boldsymbol{\theta}_t^* \rangle - \langle \mathbf{a}^{(t)}, \boldsymbol{\theta}_t^* \rangle \right]$$

$$= \mathop{\mathbb{E}}_{\mathbf{a}^{(t)} | \Omega_-^{(t)}} \left[ \max_{\mathbf{a} \in \mathcal{A}^{(t)}} \langle \mathbf{a}, \boldsymbol{\theta}_t^* \rangle - \max_{\mathbf{a} \in \mathcal{A}^{(t)}} \langle \mathbf{a}, \boldsymbol{\theta}_t \rangle + \langle \mathbf{a}^{(t)}, \boldsymbol{\theta}_t \rangle - \langle \mathbf{a}^{(t)}, \boldsymbol{\theta}_t^* \rangle \right]$$

$$= \mathop{\mathbb{E}}_{\mathbf{a}^{(t)}, \boldsymbol{\Theta}^{(t)} | \Omega_-^{(t)}} \left[ \Delta_t(\boldsymbol{\Theta}^{(t)}, \mathbf{a}^{(t)}) - \Delta_t^*(\boldsymbol{\Theta}^{(t)}) \right]. \tag{22}$$

where the first equality follows from the fact that action $\mathbf{a}^{(t)}$ maximizes $\langle \mathbf{a}, \boldsymbol{\theta}_t \rangle$ and thus $\max_{\mathbf{a} \in \mathcal{A}^{(t)}} \langle \mathbf{a}, \boldsymbol{\theta}_t \rangle = \langle \mathbf{a}^{(t)}, \boldsymbol{\theta}_t \rangle$. By combining (21) and (22), we obtain the desired result. $\square$

The following lemma shows a connection between $\Delta_t$ and a potential function.

**Lemma D.4.** *Define potential function*

$$\Phi^{(t)} := \frac{1}{\beta\lambda} \log \sum_{\boldsymbol{\Theta}\in\mathcal{H}_\epsilon} \exp\left(\beta\sum_{\tau=1}^{t}\left(L^{(\tau)}(\boldsymbol{\Theta}^*) - L^{(\tau)}(\boldsymbol{\Theta})\right)\right).$$

*Let $C_t(\mathbf{a}^{(t)}) := f^{(t)}(\mathbf{a}^{(t)}) - \langle\mathbf{a}^{(t)}, \boldsymbol{\theta}_t^*\rangle$ be the misspecification of $\boldsymbol{\Theta}^*$ on action $\mathbf{a}^{(t)}$ with respect to ground truth $f^{(t)}$. For any $\epsilon \in [0,1]$, $\beta \in (0, 0.01)$ and $\lambda \in [0,1]$, under Assumptions 3.1 and 3.2, the expected increment of potential function on any round $t$ satisfies*

$$\mathbb{E}_{\mathbf{a}^{(t)}, r^{(t)}|\Omega_-^{(t)}}\left[\Phi^{(t)} - \Phi^{(t-1)}\right] \leq -\frac{0.25}{\lambda}\mathbb{E}_{\mathbf{a}^{(t)}|\Omega_-^{(t)}}\mathbb{E}_{\boldsymbol{\Theta}\sim P^{(t)}}\left[\Delta_t(\boldsymbol{\Theta}, \mathbf{a}^{(t)})^2 + 8C_t(\mathbf{a}^{(t)})\Delta_t(\boldsymbol{\Theta}, \mathbf{a}^{(t)})\right]$$

$$+ \mathbb{E}_{\boldsymbol{\Theta}\sim P^{(t)}}\Delta_t^*(\boldsymbol{\Theta}) + 4\beta\lambda + \frac{8\beta}{\lambda}(\epsilon + \zeta)^2.$$

*Proof.* the expected increment of the potential function in round $t$ conditional on the reward function and context in round $t$ satisfies:

$$\mathbb{E}_{\mathbf{a}^{(t)}, r^{(t)}|\Omega_-^{(t)}}\left[\Phi^{(t)} - \Phi^{(t-1)}\right]$$

$$= \frac{1}{\beta\lambda}\mathbb{E}_{\mathbf{a}^{(t)}, r^{(t)}|\Omega_-^{(t)}}\left[\log\frac{\sum_{\boldsymbol{\Theta}\in\mathcal{H}_\epsilon}\exp\left(\beta\sum_{\tau=1}^{t}\left(L^{(\tau)}(\boldsymbol{\Theta}^*) - L^{(\tau)}(\boldsymbol{\Theta})\right)\right)}{\sum_{\boldsymbol{\Theta}\in\mathcal{H}_\epsilon}\exp\left(\beta\sum_{\tau=1}^{t-1}\left(L^{(\tau)}(\boldsymbol{\Theta}^*) - L^{(\tau)}(\boldsymbol{\Theta})\right)\right)}\right]$$

$$= \frac{1}{\beta\lambda}\mathbb{E}_{\mathbf{a}^{(t)}, r^{(t)}|\Omega_-^{(t)}}\left[\log\frac{\sum_{\boldsymbol{\Theta}\in\mathcal{H}_\epsilon}\exp\left(-\beta\sum_{\tau=1}^{t-1}L^{(\tau)}(\boldsymbol{\Theta})\right)\cdot\exp\left(\beta\left(L^{(t)}(\boldsymbol{\Theta}^*) - L^{(t)}(\boldsymbol{\Theta})\right)\right)}{\sum_{\boldsymbol{\Theta}\in\mathcal{H}_\epsilon}\exp\left(-\beta\sum_{\tau=1}^{t-1}L^{(\tau)}(\boldsymbol{\Theta})\right)}\right]$$

$$= \frac{1}{\beta\lambda}\mathbb{E}_{\mathbf{a}^{(t)}, r^{(t)}|\Omega_-^{(t)}}\left[\log\mathbb{E}_{\boldsymbol{\Theta}\sim P^{(t)}}\exp\left(\beta\left(L^{(t)}(\boldsymbol{\Theta}^*) - L^{(t)}(\boldsymbol{\Theta})\right)\right)\right]$$

$$\leq \frac{1}{\beta\lambda}\mathbb{E}_{\mathbf{a}^{(t)}|\Omega_-^{(t)}}\left[\log\mathbb{E}_{r^{(t)}|\Omega_-^{(t)},\mathbf{a}^{(t)}}\mathbb{E}_{\boldsymbol{\Theta}\sim P^{(t)}}\exp\left(\beta\left(L^{(t)}(\boldsymbol{\Theta}^*) - L^{(t)}(\boldsymbol{\Theta})\right)\right)\right]. \tag{23}$$

where the last equality follows from the construction of $P^{(t)}$ and the last inequality follows from Jensen's inequality. Moreover, we can decompose the loss function for any policy $\boldsymbol{\Theta}$ according to

$$L^{(t)}(\boldsymbol{\Theta}) = (r^{(t)} - \langle\mathbf{a}^{(t)}, \boldsymbol{\theta}_t\rangle)^2 - \lambda\max_{\mathbf{a}\in\mathcal{A}^{(t)}}\langle\mathbf{a}, \boldsymbol{\theta}_t\rangle$$

$$= (f^{(t)}(\mathbf{a}^{(t)}) + \xi^{(t)} - \langle\mathbf{a}^{(t)}, \boldsymbol{\theta}_t\rangle)^2 - \lambda\max_{\mathbf{a}\in\mathcal{A}^{(t)}}\langle\mathbf{a}, \boldsymbol{\theta}_t\rangle$$

$$= (C_t(\mathbf{a}^{(t)}) + \xi^{(t)} - \Delta_t(\boldsymbol{\Theta}, \mathbf{a}^{(t)}))^2 - \lambda\max_{\mathbf{a}\in\mathcal{A}^{(t)}}\langle\mathbf{a}, \boldsymbol{\theta}_t\rangle, \tag{24}$$

where the first equality follows from the definition $r^{(t)} = f^{(t)}(\mathbf{a}^{(t)}) + \xi^{(t)}$ and the second equality follows from the definition of $C_t$. For the optimal policy $\boldsymbol{\Theta}^*$ in $\epsilon$-net, one can see that

$$L^{(t)}(\boldsymbol{\Theta}^*) = (C_t(\mathbf{a}^{(t)}) + \xi^{(t)})^2 - \lambda\max_{\mathbf{a}\in\mathcal{A}^{(t)}}\langle\mathbf{a}, \boldsymbol{\theta}_t^*\rangle. \tag{25}$$

Combining (24) and (25), the terms in the exponent can be computed by

$$\beta(L^{(t)}(\boldsymbol{\Theta}^*) - L^{(t)}(\boldsymbol{\Theta}))$$
$$= -2\beta C_t(\mathbf{a}^{(t)})\Delta_t(\boldsymbol{\Theta}, \mathbf{a}^{(t)}) - 2\beta\xi^{(t)}\Delta_t(\boldsymbol{\Theta}, \mathbf{a}^{(t)}) - \beta\Delta_t(\boldsymbol{\Theta}, \mathbf{a}^{(t)})^2 + \beta\lambda\Delta_t^*(\boldsymbol{\Theta}). \tag{26}$$

Since $\xi^{(t)}$ is zero mean random variables with range 2, according to Hoeffding's lemma, we have

$$\mathbb{E}_{r^{(t)}|\Omega_-^{(t)},a^{(t)}}\left[\exp(-2\beta\xi^{(t)}\Delta_t(\boldsymbol{\Theta}, \mathbf{a}^{(t)}))\right] \leq \exp\left(2\beta^2\Delta_t(\boldsymbol{\Theta}, \mathbf{a}^{(t)})^2\right). \tag{27}$$

In case that $\beta \in (0, 0.01]$, we have $2\beta^2 - \beta \le -0.5\beta$. Plugging (26) and (27) into (23) gives

$$\mathbb{E}_{\mathbf{a}^{(t)}, r^{(t)} | \Omega_-^{(t)}} \left[ \Phi^{(t)} - \Phi^{(t-1)} \right]$$

$$\le \frac{1}{\beta\lambda} \mathbb{E}_{a^{(t)} | \Omega_-^{(t)}} \left[ \log \mathbb{E}_{\boldsymbol{\Theta} \sim P^{(t)}} \exp \left( -2\beta C_t(\mathbf{a}^{(t)}) \Delta_t(\boldsymbol{\Theta}, \mathbf{a}^{(t)}) - 0.5\beta \Delta_t(\boldsymbol{\Theta}, \mathbf{a}^{(t)})^2 + \beta\lambda \Delta_t^*(\boldsymbol{\Theta}) \right) \right].$$

According to Cauchy-Schwarz inequality, we can decompose RHS by

$$\mathbb{E}_{\mathbf{a}^{(t)}, r^{(t)} | \Omega_-^{(t)}} \left[ \Phi^{(t)} - \Phi^{(t-1)} \right] \le \underbrace{\frac{0.5}{\beta\lambda} \mathbb{E}_{a^{(t)} | \Omega_-^{(t)}} \left[ \log \mathbb{E}_{\boldsymbol{\Theta} \sim P^{(t)}} \exp \left( -4\beta C_t(\mathbf{a}^{(t)}) \Delta_t(\boldsymbol{\Theta}, \mathbf{a}^{(t)}) \right) \right]}_{I_1}$$

$$+ \underbrace{\frac{0.25}{\beta\lambda} \mathbb{E}_{a^{(t)} | \Omega_-^{(t)}} \left[ \log \mathbb{E}_{\boldsymbol{\Theta} \sim P^{(t)}} \exp \left( -2\beta \Delta_t(\boldsymbol{\Theta}, \mathbf{a}^{(t)})^2 \right) \right]}_{I_2}$$

$$+ \underbrace{\frac{0.25}{\beta\lambda} \mathbb{E}_{a^{(t)} | \Omega_-^{(t)}} \left[ \log \mathbb{E}_{\boldsymbol{\Theta} \sim P^{(t)}} \exp \left( 4\beta\lambda \Delta_t^*(\boldsymbol{\Theta}) \right) \right]}_{I_3}. \tag{28}$$

For the first term,

$$I_1 \le \frac{0.5}{\beta\lambda} \mathbb{E}_{\mathbf{a}^{(t)} | \Omega_-^{(t)}} \left[ \log \mathbb{E}_{\boldsymbol{\Theta} \sim P^{(t)}} \left[ 1 - 4\beta C_t(\mathbf{a}^{(t)}) \Delta_t(\boldsymbol{\Theta}, \mathbf{a}^{(t)}) + 16\beta^2(\epsilon + \zeta)^2 \right] \right]$$

$$\le -\frac{0.25}{\lambda} \mathbb{E}_{\mathbf{a}^{(t)} | \Omega_-^{(t)}} \mathbb{E}_{\boldsymbol{\Theta} \sim P^{(t)}} 8C_t(\mathbf{a}^{(t)}) \Delta_t(\boldsymbol{\Theta}, \mathbf{a}^{(t)}) + \frac{8\beta}{\lambda}(\epsilon + \zeta)^2 \tag{29}$$

where the first inequality follows from $e^x \le 1 + x + t^2$ for $|x| \le t \le 1$ with $|C_t(\mathbf{a}^{(t)})| \le \epsilon + \zeta \le 2$, $|\Delta_t(\boldsymbol{\Theta}, \mathbf{a}^{(t)})| \le 2$ and $\beta \in (0, 0.01]$, and the second inequality follows from $\log(1 + x) \le x$ for $x \ge -1$.

For the second term,

$$I_2 \le \frac{0.25}{\beta\lambda} \mathbb{E}_{\mathbf{a}^{(t)} | \Omega_-^{(t)}} \left[ \log \mathbb{E}_{\boldsymbol{\Theta} \sim P^{(t)}} \left[ 1 - \beta \Delta_t(\boldsymbol{\Theta}, \mathbf{a}^{(t)})^2 \right] \right]$$

$$\le -\frac{0.25}{\lambda} \mathbb{E}_{\mathbf{a}^{(t)} | \Omega_-^{(t)}} \mathbb{E}_{\boldsymbol{\Theta} \sim P^{(t)}} \Delta_t(\boldsymbol{\Theta}, \mathbf{a}^{(t)})^2, \tag{30}$$

where the first inequality follows from $e^{-x} \le 1 - 0.5x$ for $x \in [0, 1]$ with $|\Delta_t(\boldsymbol{\Theta}, \mathbf{a}^{(t)})| \le 2$ and $\beta \in (0, 0.01]$, and the second inequality follows from $\log(1 - x) \le -x$ for all $x \le 1$.

For the third term,

$$I_3 \le \frac{0.25}{\beta\lambda} \mathbb{E}_{\mathbf{a}^{(t)} | \Omega_-^{(t)}} \left[ \log \mathbb{E}_{\boldsymbol{\Theta} \sim P^{(t)}} \left[ 1 + 4\beta\lambda \Delta_t^*(\boldsymbol{\Theta}) + 16\beta^2\lambda^2 \right] \right]$$

$$\le \mathbb{E}_{\boldsymbol{\Theta} \sim P^{(t)}} \Delta_t^*(\boldsymbol{\Theta}) + 4\beta\lambda, \tag{31}$$

where the first inequality follows from $e^x \le 1 + x + t^2$ for $|x| \le t \le 1$ with $|\Delta_t^*(\boldsymbol{\Theta})| \le 2$, $\beta \in (0, 0.01]$ and $\lambda \in [0, 1]$, and the second inequality follows from $\log(1 + x) \le x$ for $x \ge -1$. Combining (29), (30) and (31) with (28), we have

$$\mathbb{E}_{\mathbf{a}^{(t)}, r^{(t)} | \Omega_-^{(t)}} \left[ \Phi^{(t)} - \Phi^{(t-1)} \right] \le -\frac{0.25}{\lambda} \mathbb{E}_{\mathbf{a}^{(t)} | \Omega_-^{(t)}} \mathbb{E}_{\boldsymbol{\Theta} \sim P^{(t)}} \left[ \Delta_t(\boldsymbol{\Theta}, \mathbf{a}^{(t)})^2 + 8C_t(\mathbf{a}^{(t)}) \Delta_t(\boldsymbol{\Theta}, \mathbf{a}^{(t)}) \right]$$

$$+ \mathbb{E}_{\boldsymbol{\Theta} \sim P^{(t)}} \Delta_t^*(\boldsymbol{\Theta}) + 4\beta\lambda + \frac{8\beta}{\lambda}(\epsilon + \zeta)^2.$$

$$\square$$

The following Lemma upper bounds the total change of the potential:

**Lemma D.5.** *Under the settings of Lemma D.4, we have*

$$\Phi^{(0)} - \Phi^{(T)} \le \frac{1}{\beta\lambda} \log N(\mathcal{H}, \epsilon, \|\cdot\|_{2,\infty}).$$

*Proof.* The statement can be proved by combining

$$\Phi^{(0)} = \frac{1}{\beta\lambda}\Big[\log \sum_{\boldsymbol{\Theta}\in\mathcal{H}_\epsilon} \exp(0)\Big] = \frac{1}{\beta\lambda}\log|\mathcal{H}_\epsilon|,$$

and

$$\Phi^{(T)} \ge \frac{1}{\beta\lambda}\mathop{\mathbb{E}}_{\Omega^{(T)}}\left[\log \max_{\boldsymbol{\Theta}\in\mathcal{H}_\epsilon} \exp\Big(\beta\sum_{\tau=1}^{T}(-L^{(\tau)}(\boldsymbol{\Theta}) + L^{(\tau)}(\boldsymbol{\Theta}^*))\Big)\right] \ge 0,$$

where the second inequality follows from $\boldsymbol{\Theta}^* \in \mathcal{H}_\epsilon$. $\qquad\square$

Note the chosen policy $\boldsymbol{\Theta}^{(t)}$ in Lemma D.3 is correlated with the selected action $\mathbf{a}^{(t)}$ while the random policy $\boldsymbol{\Theta}$ in Lemma D.4 is not. The following lemma presents a connection between these two notions.

**Lemma D.6.** *Let $C_t : \mathbb{R}^d \to \mathbb{R}$ be a function such that $|C_t(\mathbf{a})| \le \epsilon + \zeta$ for any $\mathbf{a} \in \mathcal{A}$. Then,*

$$\mathop{\mathbb{E}}_{\boldsymbol{\Theta}^{(t)}|\Omega_-^{(t)}} \Delta_t^*(\boldsymbol{\Theta}^{(t)}) = \mathop{\mathbb{E}}_{\boldsymbol{\Theta}\sim P^{(t)}} \Delta_t^*(\boldsymbol{\Theta}),$$

*and also*

$$\mathop{\mathbb{E}}_{a^{(t)},\boldsymbol{\Theta}^{(t)}|\Omega_-^{(t)}} \Delta_t(\boldsymbol{\Theta}^{(t)}, \mathbf{a}^{(t)}) \le \frac{0.25}{\lambda} \mathop{\mathbb{E}}_{\mathbf{a}^{(t)}|\Omega_-^{(t)}} \mathop{\mathbb{E}}_{\boldsymbol{\Theta}\sim P^{(t)}} \left[\Delta_t(\boldsymbol{\Theta}, \mathbf{a}^{(t)})^2 + 8C_t(\mathbf{a}^{(t)})\Delta_t(\boldsymbol{\Theta}, \mathbf{a}^{(t)})\right]$$

$$+ \lambda(d+1) + \frac{4}{\lambda}(\epsilon+\zeta)^2 + 4(\epsilon+\zeta).$$

*Proof.* According to the construction of $P^{(t)}$, the conditional distribution $\boldsymbol{\Theta}^{(t)}|\Omega_-^{(t)}$ is identical to distribution $\boldsymbol{\Theta} \sim P^{(t)}$. This directly gives

$$\mathop{\mathbb{E}}_{\boldsymbol{\Theta}^{(t)}|\Omega_-^{(t)}} \Delta_t^*(\boldsymbol{\Theta}^{(t)}) = \mathop{\mathbb{E}}_{\boldsymbol{\Theta}\sim P^{(t)}} \Delta_t^*(\boldsymbol{\Theta}).$$

Moreover, apply Lemma D.2 to joint distribution $(\mathbf{a}^{(t)}, \boldsymbol{\theta}_t^{(t)} - \boldsymbol{\theta}_t^*)$ conditional on $\Omega_-^{(t)}$, we have

$$\mathop{\mathbb{E}}_{a^{(t)},\boldsymbol{\theta}_t^{(t)}|\Omega_-^{(t)}} \langle \mathbf{a}^{(t)}, \boldsymbol{\theta}_t^{(t)} - \boldsymbol{\theta}_t^* \rangle \le \frac{0.25}{\lambda} \mathop{\mathbb{E}}_{a^{(t)}|\Omega_-^{(t)}} \mathop{\mathbb{E}}_{\boldsymbol{\theta}_t\sim P^{(t)}} \left[\langle \mathbf{a}^{(t)}, \boldsymbol{\theta}_t - \boldsymbol{\theta}_t^*\rangle^2 + 8C_t(\mathbf{a}^{(t)})\langle \mathbf{a}^{(t)}, \boldsymbol{\theta}_t - \boldsymbol{\theta}_t^*\rangle\right]$$

$$+ \lambda(d+1) + \frac{4}{\lambda}(\epsilon+\zeta)^2 + 4(\epsilon+\zeta).$$

The desired statement can be obtained using the definition of $\Delta_t$. $\qquad\square$

Now, we are ready to prove Theorem 5.1:

*Proof of Theorem 5.1.* We choose the parameters according to

$$\lambda := \sqrt{\frac{\log|\mathcal{H}_\epsilon| + (\epsilon+\zeta)^2 T}{dT}} = \mathcal{O}(1/d), \beta := 0.01. \tag{32}$$

Denote

$$\mathcal{X}_t := \frac{0.25}{\lambda} \mathop{\mathbb{E}}_{\mathbf{a}^{(t)}|\Omega_-^{(t)}} \mathop{\mathbb{E}}_{\boldsymbol{\Theta}\sim P^{(t)}} \left[\Delta_t(\boldsymbol{\Theta}, \mathbf{a}^{(t)})^2 + 8C_t(\mathbf{a}^{(t)})\Delta_t(\boldsymbol{\Theta}, \mathbf{a}^{(t)})\right] - \mathop{\mathbb{E}}_{\boldsymbol{\Theta}\sim P^{(t)}} \Delta_t^*(\boldsymbol{\Theta}).$$

According to Lemma D.6, we have

$$\mathop{\mathbb{E}}_{\mathbf{a}^{(t)},\boldsymbol{\Theta}^{(t)}|\Omega_-^{(t)}} \left[\Delta_t(\boldsymbol{\Theta}^{(t)}, \mathbf{a}^{(t)}) - \Delta_t^*(\boldsymbol{\Theta}^{(t)})\right] \le \mathcal{X}_t + \lambda(d+1) + \frac{4}{\lambda}(\epsilon+\zeta)^2 + 4(\epsilon+\zeta). \tag{33}$$

According to Lemma D.4, we have

$$\underset{\mathbf{a}^{(t)}, r^{(t)}|\Omega_-^{(t)}}{\mathbb{E}}\left[\Phi^{(t)} - \Phi^{(t-1)}\right] \leq -\mathcal{X}_t + 4\beta\lambda + \frac{8\beta}{\lambda}(\epsilon + \zeta)^2. \tag{34}$$

Combining (33) and (34) gives

$$\underset{\mathbf{a}^{(t)}, \mathbf{\Theta}^{(t)}|\Omega_-^{(t)}}{\mathbb{E}}\left[\Delta_t(\mathbf{\Theta}^{(t)}, \mathbf{a}^{(t)}) - \Delta_t^*(\mathbf{\Theta}^{(t)})\right]$$

$$\leq \underset{\mathbf{a}^{(t)}, r^{(t)}|\Omega_-^{(t)}}{\mathbb{E}}\left[\Phi^{(t-1)} - \Phi^{(t)}\right] + \lambda(d+1+4\beta) + \frac{4+8\beta}{\lambda}(\epsilon + \zeta)^2 + 4(\epsilon + \zeta). \tag{35}$$

Moreover, we have

$$\mathbb{E}[\text{Regret}(T)]$$

$$= \sum_{t=1}^{T} \underset{\mathbf{a}^{(t)}|\Omega_-^{(t)}}{\mathbb{E}}\left[f^{(t)}(\mathbf{a}_*^{(t)}) - f^{(t)}(\mathbf{a}^{(t)})\right]$$

$$\leq \sum_{t=1}^{T} \underset{\mathbf{a}^{(t)}, \mathbf{\Theta}^{(t)}|\Omega_-^{(t)}}{\mathbb{E}}\left[\Delta_t(\mathbf{\Theta}^{(t)}, \mathbf{a}^{(t)}) - \Delta_t^*(\mathbf{\Theta}^{(t)})\right] + 2(\epsilon + \zeta)T$$

$$\leq \sum_{t=1}^{T} \underset{\mathbf{a}^{(t)}, r^{(t)}|\Omega_-^{(t)}}{\mathbb{E}}\left[\Phi^{(t-1)} - \Phi^{(t)}\right] + \lambda(d+1+4\beta)T + \frac{4+8\beta}{\lambda}(\epsilon + \zeta)^2 T + 6(\epsilon + \zeta)T$$

$$\leq \frac{1}{\beta\lambda}\log|\mathcal{H}_\epsilon| + \lambda(d+1+4\beta)T + \frac{4+8\beta}{\lambda}(\epsilon + \zeta)^2 T + 6(\epsilon + \zeta)T,$$

where the first inequality follows from Lemma D.3, the second inequality follows from (35), and the last inequality follows from Lemma D.5. With the parameter defined in (32), we conclude that

$$\mathbb{E}[\text{Regret}(T)] \leq O\left(\sqrt{dT\log|\mathcal{H}_\epsilon|} + T\sqrt{d}(\epsilon + \zeta)\right).$$

$\square$

# E  PROOF OF COVERING NUMBER FROM SECTION 5.1

In this section, we prove covering number bounds for some specific linear bandit settings. We first introduce a classic result on the covering number of the unit ball.

**Lemma E.1** (Lemma 5.2 in Vershynin (2012), Restated). *The unit Euclidean ball $B^d = \{\mathbf{x} \in \mathbb{R}^d : \|\mathbf{x}\|_2 \leq 1\}$ equipped with the Euclidean metric satisfies for every $\epsilon > 0$ that*

$$N(B^d, \epsilon, \|\cdot\|_2) \leq (1 + 2\epsilon^{-1})^d.$$

With this lemma, we can directly prove the covering number of stationary linear bandits.

**Lemma 5.1** (Covering Number for Misspecified Linear Bandits). *Under Assumptions 3.1 and 3.3, the metric entropy of linear policies satisfies*

$$\log N(\mathcal{H}, \epsilon, \|\cdot\|_{2,\infty}) \leq \mathcal{O}(d\log\epsilon^{-1}).$$

*Proof.* We will prove this lemma by constructing an $\epsilon$-net for $\mathcal{H}$. Let $B_\epsilon^d$ be the minimum $\epsilon$-net of unit Euclidean ball $B^d$ in $L_2$ distance. Let $\mathcal{H}_\epsilon$ be the set that contains every matrix $[\widetilde{\boldsymbol{\theta}}, \cdots, \widetilde{\boldsymbol{\theta}}]$ for all $\widetilde{\boldsymbol{\theta}} \in B_\epsilon^d$. According to Lemma E.1, we have $|\mathcal{H}_\epsilon| = |B_\epsilon^d| \leq (1 + 2\epsilon^{-1})^d$.

We then show $\mathcal{H}_\epsilon$ is an $\epsilon$-net of $\mathcal{H}$ under Assumption 3.3: For every linear policy $[\boldsymbol{\theta}, \cdots, \boldsymbol{\theta}]$, there exists $\widetilde{\boldsymbol{\theta}} \in B_\epsilon^d$ such that $\|\boldsymbol{\theta} - \widetilde{\boldsymbol{\theta}}\|_2 \leq \epsilon$ according to definition of $B_\epsilon^d$. This shows $[\widetilde{\boldsymbol{\theta}}, \cdots, \widetilde{\boldsymbol{\theta}}] \in \mathcal{H}_\epsilon$ is an $\epsilon$-approximation policy of $[\boldsymbol{\theta}, \cdots, \boldsymbol{\theta}]$ which implies $\mathcal{H}_\epsilon$ is indeed an $\epsilon$-net of $\mathcal{H}$.

As a result, we can bound the metric entropy by

$$\log N(\mathcal{H}, \epsilon, \|\cdot\|_{2,\infty}) \leq \log|\mathcal{H}_\epsilon| = \mathcal{O}(d\log\epsilon^{-1}).$$

$\square$

One can naturally extend the proof to non-stationary linear bandits with bounded switches:

**Lemma 5.2** (Covering Number for Non-Stationary Linear Bandits with Bounded Switches). *Under Assumptions 3.1 and 3.4, the metric entropy of linear policies satisfies*

$$\log N(\mathcal{H}, \epsilon, \|\cdot\|_{2,\infty}) \leq \mathcal{O}(dS \log \epsilon^{-1} + S \log T).$$

*Proof.* We will prove this lemma by constructing an $\epsilon$-net for $\mathcal{H}$. Let $B_\epsilon^d$ be the minimum $\epsilon$-net of unit Euclidean ball $B^d$ in $L_2$ distance. We build a set $\mathcal{H}_\epsilon$ by mapping any sequence $(t_1, \cdots, t_S)$ with $1 < t_1 < \cdots < t_S \leq T$ and $S + 1$ vectors $\widetilde{\boldsymbol{\theta}}_0, \cdots, \widetilde{\boldsymbol{\theta}}_S \in B_\epsilon^d$ to the matrix $[\widetilde{\boldsymbol{\theta}}_0, \cdots, \widetilde{\boldsymbol{\theta}}_0, \widetilde{\boldsymbol{\theta}}_1, \cdots, \widetilde{\boldsymbol{\theta}}_1, \cdots, \widetilde{\boldsymbol{\theta}}_S]$ where the first $\widetilde{\boldsymbol{\theta}}_i$ occurs at index $t_i$. $t_i$ describes the round of $i$-th switch and $\widetilde{\boldsymbol{\theta}}_i$ characterize the parameter after the switch. The size of $\mathcal{H}_\epsilon$ can be bounded by

$$|\mathcal{H}_\epsilon| \leq \binom{T}{S} \cdot |B_\epsilon^d|^{S+1} \leq T^S \cdot (1 + 2\epsilon^{-1})^{d(S+1)},$$

where the first term is the number of possible $(t_i)$ and the second term is the number of possible $\widetilde{\boldsymbol{\theta}}_i$. We then show $\mathcal{H}_\epsilon$ is an $\epsilon$-net of $\mathcal{H}$ under under Assumption 3.4: Every linear policy in $\mathcal{H}$ can also be written in form $[\boldsymbol{\theta}_0, \cdots, \boldsymbol{\theta}_0, \boldsymbol{\theta}_1, \cdots, \boldsymbol{\theta}_1, \cdots, \boldsymbol{\theta}_S]$ which can be covered by some $[\widetilde{\boldsymbol{\theta}}_0, \cdots, \widetilde{\boldsymbol{\theta}}_0, \widetilde{\boldsymbol{\theta}}_1, \cdots, \widetilde{\boldsymbol{\theta}}_1, \cdots, \widetilde{\boldsymbol{\theta}}_S]$ with the same switch locations and corresponding parameters. According to the definition of $B_\epsilon^d$, $\|\boldsymbol{\theta}_i - \widetilde{\boldsymbol{\theta}}_i\| \leq \epsilon$ for every $i$ and thus $\mathcal{H}_\epsilon$ is an $\epsilon$-net of $\mathcal{H}$.

As a result, we can bound the metric entropy by

$$\log N(\mathcal{H}, \epsilon, \|\cdot\|_{2,\infty}) \leq \log |\mathcal{H}_\epsilon| = \mathcal{O}(dS \log \epsilon^{-1} + S \log T).$$

$\square$

For non-stationary linear bandits with bounded path length, we will show the $\epsilon$-net for some bounded switches instances can be transformed into a desired covering:

**Lemma 5.3** (Covering Number for Non-Stationary Linear Bandits with Bounded Path Length). *Under Assumptions 3.1 and 3.5, the metric entropy of linear policies satisfies*

$$\log N(\mathcal{H}, \epsilon, \|\cdot\|_{2,\infty}) \leq \mathcal{O}(d \log \epsilon^{-1} + dP\epsilon^{-1} \log \epsilon^{-1} + P\epsilon^{-1} \log T).$$

*Proof.* Let $\widehat{\mathcal{H}}_{\epsilon/2}$ be the $\epsilon/2$-net covering of non-stationary bandits with no more than $2P\epsilon^{-1}$ (see Assumption 3.4). We will show $\widehat{\mathcal{H}}_{\epsilon/2}$ is an $\epsilon$-net covering of non-stationary bandits with path length no more than $P$:

For some linear policy $\boldsymbol{\Theta} = [\boldsymbol{\theta}_1, \boldsymbol{\theta}_2, \cdots, \boldsymbol{\theta}_T] \in \mathcal{H}$ such that $\sum_{t=1}^T \|\boldsymbol{\theta}_t - \boldsymbol{\theta}_{t-1}\| \leq P$, consider a sequence $1 = t_0 < t_1 < \cdots < t_M \leq T$ where $t_i$ is the minimal index such that $\|\boldsymbol{\theta}_{t_i} - \boldsymbol{\theta}_{t_{i-1}}\| > \epsilon/2$ for all $i \geq 1$. Denote $\widehat{\boldsymbol{\Theta}} = [\boldsymbol{\theta}_1, \cdots, \boldsymbol{\theta}_1, \boldsymbol{\theta}_{t_1}, \cdots, \boldsymbol{\theta}_{t_1}, \cdots, \boldsymbol{\theta}_{t_M}]$ as the linear policy, which quantizes the parameter drift in which the first $\boldsymbol{\theta}_{t_i}$ occurs at index $t_i$. Since the number of switch of $\widehat{\boldsymbol{\Theta}}$ is bounded by $M \leq P/(\epsilon/2) \leq 2P\epsilon^{-1}$, the $\epsilon/2$-net $\widehat{\mathcal{H}}_{\epsilon/2}$ contains an $\epsilon/2$ approximated policy of $\widehat{\boldsymbol{\Theta}}$. Denote the policy as $\widetilde{\boldsymbol{\Theta}}$.

Since deviation of each single round policy in the same slice is no more than $\epsilon/2$, we have $\|\widehat{\boldsymbol{\Theta}} - \boldsymbol{\Theta}\|_{2,\infty} \leq \epsilon/2$. According to the definition of $\epsilon/2$-net, we also have $\|\widetilde{\boldsymbol{\Theta}} - \widehat{\boldsymbol{\Theta}}\|_{2,\infty} \leq \epsilon/2$. According to triangle inequality, we have $\|\widetilde{\boldsymbol{\Theta}} - \boldsymbol{\Theta}\|_{2,\infty} \leq \epsilon$. This shows that $\boldsymbol{\Theta}$ always has an $\epsilon$-approximation within $\widehat{\mathcal{H}}_{\epsilon/2}$ and thus $\widehat{\mathcal{H}}_{\epsilon/2}$ is indeed an $\epsilon$-net of $\mathcal{H}$.

As a result, we can bound the metric entropy using Lemma 5.2

$$\log N(\mathcal{H}, \epsilon, \|\cdot\|_{2,\infty}) \leq \log |\widehat{\mathcal{H}}_{\epsilon/2}| \leq \mathcal{O}(d \log \epsilon^{-1} + dP\epsilon^{-1} \log \epsilon^{-1} + P\epsilon^{-1} \log T),$$

where the extra $d \log \epsilon^{-1}$ term follows from the special case $S = 2P\epsilon^{-1} < 1$. $\square$

Now we consider the lifelong linear bandits. Its covering number bound is essentially the covering number of low-rank matrices.

**Lemma 5.4** (Covering Number for Lifelong Bandits with Shared Representation). *Under Assumptions 3.1 and 3.6, the metric entropy of linear policies satisfies*

$$\log N(\mathcal{H}, \epsilon, \|\cdot\|_{2,\infty}) \leq \mathcal{O}(dk \log \epsilon^{-1} + kM \log \epsilon^{-1}).$$

*Proof.* We will prove this lemma by constructing an $\epsilon$-net for $\mathcal{H}$. This is done by constructing an $\epsilon/2$-net for all possible feature extractor $\mathbf{B}$ and an $\epsilon/2$-net for all possible tasks specific vectors $\{\mathbf{w}_i\}_{i=1}^M$. In specific, we build a set $\mathcal{B}_{\epsilon/2}$ such that there exists $\widetilde{\mathbf{B}} \in \mathcal{B}_{\epsilon/2}$ such that $\|\mathbf{B} - \widetilde{\mathbf{B}}\|_2 \leq \epsilon/2$ and $\mathcal{W}_{\epsilon/2}$ such that there exists $\{\widetilde{\mathbf{w}}_i\}_{i=1}^M \in \mathcal{W}_{\epsilon/2}$ such that $\|\mathbf{w}_i - \widetilde{\mathbf{w}}_i\|_2 \leq \epsilon/2$ for all $i$. We generate the $\epsilon$-net by mapping every pairs of $\widetilde{\mathbf{B}}$ and $\{\widetilde{\mathbf{w}}_i\}_{i=1}^M$ to matrix $[\widetilde{\boldsymbol{\theta}}^{(1)}, \cdots, \widetilde{\boldsymbol{\theta}}^{(T)}]$ where $\widetilde{\boldsymbol{\theta}}^{(t)} = \mathbf{B}\widetilde{\mathbf{w}}_{m^{(t)}}$.

In any round with $\boldsymbol{\theta}^{(t)} = \mathbf{B}\mathbf{w}_{m^{(t)}}$, the vector $\widetilde{\boldsymbol{\theta}}^{(t)} = \widetilde{\mathbf{B}}\widetilde{\mathbf{w}}_{m^{(t)}}$ with corresponding $\widetilde{\mathbf{B}}$ and $\widetilde{\mathbf{w}}_{m^{(t)}}$ in $\epsilon/2$-net satisfies

$$
\begin{aligned}
\|\boldsymbol{\theta}^{(t)} - \widetilde{\boldsymbol{\theta}}^{(t)}\|_2 &= \|\mathbf{B}\mathbf{w}_{m^{(t)}} - \widetilde{\mathbf{B}}\widetilde{\mathbf{w}}_{m^{(t)}}\|_2 \\
&\leq \|\mathbf{B}\mathbf{w}_{m^{(t)}} - \mathbf{B}\widetilde{\mathbf{w}}_{m^{(t)}}\|_2 + \|\mathbf{B}\widetilde{\mathbf{w}}_{m^{(t)}} - \widetilde{\mathbf{B}}\widetilde{\mathbf{w}}_{m^{(t)}}\|_2 \\
&\leq \|\mathbf{B}\|_2\|\mathbf{w}_{m^{(t)}} - \widetilde{\mathbf{w}}_{m^{(t)}}\|_2 + \|\mathbf{B} - \widetilde{\mathbf{B}}\|_2\|\widetilde{\mathbf{w}}_{m^{(t)}}\|_2 \\
&\leq 1 \cdot \epsilon/2 + 1 \cdot \epsilon/2 \leq \epsilon,
\end{aligned}
$$

where the second inequality follows from triangle inequality and the third inequality holds since $\mathbf{B}$ is orthogonal matrix. This shows any task can be $\epsilon$-approximated and the generated set is indeed an $\epsilon$-net. As a result, with Lemma E.1, we conclude that

$$
\log N(\mathcal{H}, \epsilon, \|\cdot\|_{2,\infty}) \leq \log |\mathcal{B}_{\epsilon/2}| + \log |\mathcal{W}_{\epsilon/2}| \leq \mathcal{O}(dk \log \epsilon^{-1} + kM \log \epsilon^{-1}).
$$

$\square$

## F  LOWER BOUNDS IN SECTION 5.1

In this section, we give regret lower bounds for some specific linear bandit settings. We first present some existing result in the literature. It is easy to check the corresponding hard instance satisfies our formulation.

**Lemma F.1.** *(Theorem 24.1 in Lattimore & Szepesvári (2020), Restated) Under Assumptions 3.1, 3.2 and 3.3, for any algorithm, there exists a bandit instance for which*

$$
\mathbb{E}[\text{Regret}(T)] \geq \Omega\left(d\sqrt{T}\right).
$$

**Lemma F.2.** *(Theorem 1 in Cheung et al. (2019), Restated) Under Assumptions 3.1, 3.2 and 3.5, for any algorithm, there exists a bandit instance for which*

$$
\mathbb{E}[\text{Regret}(T)] \geq \Omega\left(d^{\frac{2}{3}} T^{\frac{2}{3}} P^{\frac{1}{3}}\right).
$$

**Lemma F.3.** *(Theorem 4 in Yang et al. (2020), Restated) Under Assumptions 3.1, 3.2 and 3.6, for any algorithm, there exists a bandit instance for which*

$$
\mathbb{E}[\text{Regret}(T)] \geq \Omega\left(d\sqrt{kT} + k\sqrt{MT}\right).
$$

**Lemma F.4.** *(Theorem F.1 in Lattimore & Szepesvari (2020), Restated) Under Assumptions 3.1 and 3.2, for any algorithm, there exists a bandit instance for which*

$$
\mathbb{E}[\text{Regret}(T)] \geq \Omega\left(T\sqrt{d/\log T}\zeta\right).
$$

**Remark F.1.** *In Lattimore & Szepesvari (2020), they proved a lower bound in the order of $\Omega(\min(T, K)\sqrt{d/\log(K)}\zeta)$, where $K$ is the number of actions. Here we choose $K = T$ for simplicity.*

We further prove a lower bound for non-stationary linear bandits with bounded switches.

**Lemma F.5.** *Under Assumptions 3.1, 3.2 and 3.4, for any algorithm, there exists a bandit instance for which*

$$
\mathbb{E}[\text{Regret}(T)] \geq \Omega\left(d\sqrt{ST}\right).
$$

*Proof.* Without loss of generality, assume $T$ is divided by $S$. Consider a bandit instance where the switch occurs at round $T/S, 2T/S, \cdots$. Note the reward functions after different switches are independent. Thus, according to Lemma F.1, any algorithm suffers $\Omega(d\sqrt{T/S})$ regret in any interval between two switches. Since there are $S$ switches, for any algorithm, there exists a bandit instance for which $\mathbb{E}[\text{Regret}(T)] \geq \Omega(d\sqrt{ST})$. $\square$

The lower bounds result in Section 5.1 can be obtained by combining above lemmas:

- Combining Lemma F.1 and Lemma F.4 gives Proposition 5.1.
- Combining Lemma F.5 and Lemma F.4 gives Proposition 5.2.
- Combining Lemma F.2, Lemma F.1, and Lemma F.4 gives Proposition 5.3.
- Combining Lemma F.3 and Lemma F.4 gives Proposition 5.4.

