# OpenReview forum: "The Power of Feel-Good Thompson Sampling: A Unified Framework for Linear Bandits"
_ICLR.cc/2023/Conference — Submitted to ICLR 2023_

### Official Review · Reviewer_iGav · 2022-10-24

**Confidence:** 4
**Correctness:** 4
**Technical Novelty And Significance:** 3
**Empirical Novelty And Significance:** Not applicable
**Recommendation:** 6

**Clarity, Quality, Novelty And Reproducibility:**

This work does not include experiments, and I think the quality would be improved if the weaknesses mentioned above could be fixed.

**Strength And Weaknesses:**

Strength:
1. The paper is generally easy to follow and well-organized.
2. The literature review is detailed.

Weaknesses:
1. In section 4.3, the comparison with FGTS mentions that "the original FGTS samples over models (or equivalently, reward functions) rather than policies". However, a more vital question is: what is the corresponding analytical challenge?
2. Since Algorithm 1 is not computationally efficient and even hard to implement, what are the possible solutions? Any thoughts?
3. In section 5, when discussing the results in the four variations of linear bandits, the contribution of this work would be clearer if the author(s) could provide a comparison between the existing bounds and those provided by this work.
4. Small typo: page 6 --- before Definition 4.2: should be "we define the approach of $\epsilon$-net and covering number."

**Summary Of The Paper:**

This work aims to provide a unified Thompson-sampling based algorithm for linear bandits. It provides nearly-matching upper and lower bounds, and also discusses the results in some specific cases: misspecified linear bandits, non-stationary linear bandits with bounded switches, non-stationary linear bandits with bounded path length, and lifelong linear bandits with shared representation.

**Summary Of The Review:**

Generally speaking, this work tries to deepen the study of feel-good Thompson-sampling algorithm proposed in Zhang (2021) and it does make some contribution to the field. I would appreciate that if the weaknesses above could be solved.


=========

After rebuttal:
Thanks for your efforts. I increased the score to 6.

---

> ### Author Response · Authors · 2022-11-15
> **Response to Reviewer iGav**
>
> Thank you for your helpful comments and suggestions! We address your questions and concerns as follows.
>
> **Q1**: What is the corresponding analytical challenge compared to FGTS?
>
> **A1**: The main challenges are:
>
> - In the original FGTS, they only sample one model to make decisions. In comparison, we need to maintain a sequence of models (except for the misspecified linear bandits). As a result, one has to get a generalized version of the decoupling lemma, i.e., Lemma D.1 in our paper, to handle the sequence of models.
>
> - Besides, to handle the probability distribution over a sequence of models, we introduced epsilon-net to measure its complexity. This requires the algorithm to handle misspecification in an explicit way. This further requires a decoupling lemma that can handle misspecification. This is achieved by Lemma D.2 in our paper.
>
> ***
>
> **Q2**: What are the possible solutions to implement Algorithm 1?
>
> **A2**: Algorithm 1 can be implemented using Langevin Monte Carlo. We present a detailed discussion on this and simulation results in Appendix B.
>
> ***
> **Q3**: “the contribution of this work would be clearer if the author(s) could provide a comparison between the existing bounds and those provided by this work.”
>
> **A3**: Thank you for your suggestion. We have added a comparison in Table 1 in the main paper.
>
> ***
> **Q4**: Small typo
>
> **A4**: Thank you for pointing out the typo in our paper. We have fixed it accordingly.

---

### Official Review · Reviewer_oQHX · 2022-10-24

**Confidence:** 3
**Correctness:** 4
**Technical Novelty And Significance:** 3
**Empirical Novelty And Significance:** Not applicable
**Recommendation:** 5

**Clarity, Quality, Novelty And Reproducibility:**

The paper is generally clear, the have made references to related work and the overall presentation of the work is good.

**Strength And Weaknesses:**

The algorithm proposes a general framework that then is able to be close to optimal for the three cases that are shown. They introduce an algorithm that is close to optimal for the lifelong case.

The main drawback is that it would be been good to see some simulations results of the methods to see if even in toy examples, these bounds hold in simulation and that also gives a good visual of the constants involved in the algorithm.

**Summary Of The Paper:**

The authors use Feel Good Thompson sampling to derive general bounds for three different linear contextual bandits 1) non-stationary bandits with S switches, 2) non-stationary bandits with path-length P and 3) lifelong bandits over M tasks. In each case they show that their general bound is close to optimal of the lower bound (nearly minimax for 1 and 2 but 2 has a bigger gap between upper and lower bounds).

**Summary Of The Review:**

Overall it is a good paper that gives a near-optimal algorithm for the lifelong learning setting.

---

> ### Author Response · Authors · 2022-11-15
> **Response to Reviewer oQHX**
>
> Thank you for your suggestion! We have revised our paper to add discussions on the implementation of the algorithm and presented corresponding simulation results in Appendix B.

---

> ### Author Response · Authors · 2022-11-21
> **Follow up with Reviewer oQHX**
>
> Dear Reviewer,
>
>
> We’d like to follow up with you to hear your feedback on our revised paper. Since you think “The main drawback is that it would be good to see some simulations results of the methods to see if even in toy examples”, we have added the detailed implementation and simulation for misspecified linear bandits and life-long linear bandits in the appendix. Please let us know if you’re satisfied with that. Thank you.
>
> Best,
> Authors

---

### Official Review · Reviewer_vpXj · 2022-10-26

**Confidence:** 3
**Correctness:** 3
**Technical Novelty And Significance:** 2
**Empirical Novelty And Significance:** Not applicable
**Recommendation:** 5

**Clarity, Quality, Novelty And Reproducibility:**

This paper is a follow-up work to Zhang (2021). It is not hard to read the paper. Since the proposed algorithm is (almost) unimplementable, there is no any empirical result.

**Strength And Weaknesses:**

The main strength of the paper is that the proposed algorithm has (near-)optimal regret bounds for several variants of linear contextual bandits.

The main weakness is that the proposed algorithm is (almost) unimplementable. I could not see a practical way to implement this algorithm.  Besides this , I have several questions and comments.
1. The paper claims resolving the respective open problem in each setting of linear contextual bandits. I think it is better to clearly describe these open problems.
2. Why $\pi$ is called (meta-)policy? To me, it is the whole model of $T$ periods.
3. It seems that the proposed algorithm does not need to know the parameters including the misspecification level, the bound of number of switches denoted by $S$ and the bound of the path length denoted by $P$, right? If my understanding is correct, could the authors elaborate more on how the analysis of the algorithm deals with these unknown parameters?
4. In terms of the algorithm design, why at each period, the algorithm samples the whole model of $T$ periods? I mean at each period $t$, why we need to obtain samples of $\theta$'s before $t$? I think this is the main step that makes the algorithm not implementable.
5. In terms of the regret bound in Theorem 5.1, how sensitive is this bound to the choices of $\lambda$ and $\beta$? The conditions of $\lambda$ and $\beta$ are in the form of $\Theta$. What are the potential ranges of the hidden constants?
6. The analysis follows and extends that in Zhang (2021) based on the notion of Decoupling Coefficient (DC). After Zhang (2021), Dylan et al. (2022) introduces a very similar notion called Decision-Estimation Coefficient (DEC) and proposes a general algorithm based on DEC. However, their algorithm's regret bound even for linear bandits is suboptimal in $d$. Could the authors elaborate why the analysis based on DC gives the optimal dependence in $d$, while that based on DEC does not?

Dylan J. Foster, Sham M. Kakade, Jian Qian, and Alexander Rakhlin, The Statistical Complexity of Interactive Decision Making, 2022




**Summary Of The Paper:**

The paper introduces a framework that covers several variants of linear contextual bandits, and proposes a variant of feel-good Thompson sampling introduced in Zhang (2021). The theoretical results show that the proposed algorithm has (near-)optimal regret bounds for various settings of linear contextual bandits.

**Summary Of The Review:**

My main concern is about the practicality of the proposed algorithm. For example, how to implement even an approximation of the proposed algorithm.

---

> ### Author Response · Authors · 2022-11-15
> **Response to Reviewer vpXj**
>
> Thank you for your constructive comments! We address your questions and concerns as follows.
>
> **Q0**: “the main weakness is that the proposed algorithm is (almost) unimplementable”
>
> **A0**: Algorithm 1 can be practically implemented with the help of Langevin Monte Carlo. In the revision, we present a detailed discussion and simulation results in Appendix B.
>
> ***
>
> **Q1**: “I think it is better to clearly describe these open problems.”
>
> **A1**: We have added descriptions of each open problem in Section 3.2.
>
> ***
>
> **Q2**: “Why $\pi$ is called  (meta-)policy? To me, it is the whole model of $T$ periods.”
>
> **A2**: Thanks for your suggestion. We have changed it to “a sequence of models”.
>
> ***
>
> **Q3**: Does the algorithm need to know the parameters including the misspecification level $\xi$, the bound of the number of switches denoted by $S$, and the bound of the path length denoted by $P$?
>
> **A3**: Yes. Since we choose $\lambda$ according to the covering number $\mathcal{N}(\mathcal{H}, \epsilon)$ which is related to all $\xi$, $S$, and $P$, the algorithm needs to know these parameters. In practice, $\lambda$ can be tuned using grid search.
>
> ***
>
> **Q4**: “Why do we need to obtain samples of $\mathbf{\theta}$ before $t$?”
>
> **A4**: In general, the bandits models we are dealing with are time-varying and can be described as a sequence of models. After $\epsilon$-net covering, the number of possible sequences of models are combinatorial. Therefore, we need to sample the whole sequence of models at each time t to find the true sequence of models. For stationary misspecified linear bandits, since the bandit model is fixed over time and there is no combinatorial structure, it is sufficient to sample one $\mathbf{\theta}$ at each time t. More details and discussions can be found in Appendix B.
>
> ***
>
> **Q5**: In terms of the regret bound in Theorem 5.1, how sensitive is this bound to the choices of $\lambda$ and $\beta$?
>
> **A5**: According to our proof in Section D, the choices of $\lambda$ and $\beta$ are not sensitive to the constant hidden in the Big-Theta notation. In particular, according to our proof of Lemma D.4, we need $\beta \in (0,0.01]$.
>
> ***
>
> **Q6**: Is the regret bound of the algorithm based on DEC suboptimal in $d$ for linear bandits?
>
> **A6**: The regret bound of the algorithm based on DEC is in the form of $\sqrt{d T \mathrm{est}}$ where $\mathrm{est}=O(d)$ for linear bandits. Therefore, the regret bound of the algorithm is also tight for linear bandits.

---

> ### Author Response · Authors · 2022-11-20
> **Follow up with Reviewer vpXj**
>
> Dear Reviewer,
>
> We’d like to follow up with you to hear your feedback on our response and revised paper. Since your main concern is “about the practicality of the proposed algorithm. For example, how to implement even an approximation of the proposed algorithm.”, we believe we have fully addressed your concern by adding the detailed implementation and experiments for misspecified linear bandits and life-long linear bandits in the appendix. We have also addressed your other questions. Please let us know if you’re satisfied with that. Thank you.
>
> Best,
> Authors.

---

> > ### Comment · Reviewer_vpXj · 2022-11-21
> > **implementation and experiments for non-stationary linear bandits?**
> >
> > Thank you for your response. Actually I was worried about the practicality of the proposed algorithm for non-stationary linear bandits when the algorithm needs to (re)sample a sequence of $T$ models at each time $t$.

---

> > > ### Author Response · Authors · 2022-11-21
> > > **Re: implementation and experiments for non-stationary linear bandits**
> > >
> > > Dear Reviewer,
> > >
> > > When applied to non-stationary linear contextual bandits with bounded path length, our algorithm is computationally expensive, as it requires to sample a $d\times T$ matrix $\boldsymbol{\Theta}$ at each round under the path length constraint. This needs to be approximated by LMC over a constraint set. So far we don’t have any experiments on this. When applied to non-stationary linear contextual bandits with bounded switches, our algorithm is computationally inefficient since it requires to sample a $d\times T$ matrix $\boldsymbol{\Theta}$ under certain combinatorial constraints.  However, we would like to emphasize that, ignoring the computational efficiency, the algorithms derived from our framework are the first nearly-minimax optimal ones for non-stationary linear contextual bandits, which respectively resolve two open problems proposed by Luo et al. (2022) on the bounded switches variants and Faury et al. (2021) on the bounded path length variants. Even though the resulting algorithms are computationally expensive or even inefficient, they are still significant progress towards optimal algorithms for non-stationary linear contextual bandits.
> > >
> > >
> > > In fact, our paper proposes a unified framework for linear contextual bandits, and we give three instantiations of our framework (i.e., misspecified linear contextual bandits, multi-task linear contextual bandits, life-long linear contextual bandits) that are both computationally efficient and statistically (near) optimal. In particular, we proposed the first practical algorithm for lifelong linear contextual bandits with nearly-optimal regret bound and a new practical algorithm for misspecified linear contextual bandits with nearly-minimax regret bound. All these algorithms are statistically and computationally efficient. They are also backed by the experiments.
> > >
> > > Best, Authors.

---

### Author Response · Authors · 2022-11-15
**To all reviewers**

Thank you for all your valuable comments. We have revised our manuscript accordingly. The major revisions are marked red in the paper. Below is a brief summary of the revisions:

1. We have included a detailed discussion on the implementation of the algorithm and provided simulation results on synthetic data for different settings in Appendix B.
2. We have added descriptions of the open problems for the variants of linear contextual bandit in Section 5.2.
3. We have added a table comparing our algorithm with existing results in various settings in Section 5.
4. We have fixed the minor issues including typos.

---

### Decision · Program_Chairs · 2023-01-20

**Decision:**

Reject

**Justification For Why Not Higher Score:**

As above

**Justification For Why Not Lower Score:**

Can't go lower.

**Metareview: Summary, Strengths And Weaknesses:**

This paper proposes a framework that encompasses several variants of linear bandits based on feel-good Thompson sampling (Zhang, 2021). The proposed regret bounds have (near-)optimal performances for linear bandits. The analysis of this paper is commendable and unifies previous analyses.

However, this paper is borderline and as the AC, I discussed it in detail with the three reviewers. There was a strong consensus that the paper has made important theoretical contributions to the bandit literature. However, all three reviewers felt that for the paper to be published in this or other conferences, it is paramount to conduct implementations and experiments for non-stationary linear bandits and compare the performance to existing algorithms. Without which, the practicality of FGTS.LP cannot be unequivocally ascertained. Release of code would further strengthen the paper and its potential impact.

**Summary Of Ac-Reviewer Meeting:**

I had a thorough email discussion with the reviewers. All strongly agreed that the paper contains strong theoretical contributions but experiments are lacking, which results in the paper being not publishable in its current form.